# Sulfur and Water Resistance of Carbon-Based Catalysts for Low-Temperature Selective Catalytic Reduction of NO$_x$: A Review

Zhenghua Shen [1,2], Shan Ren [3,*], Baoting Zhang [4], Weixin Bian [4], Xiangdong Xing [1,2,*] and Zhaoying Zheng [1,2]

1 School of Metallurgy Engineering, Xi'an University of Architecture and Technology, Xi'an 710055, China; shenzhenghhua@xauat.edu.cn (Z.S.); junkook19970901@163.com (Z.Z.)
2 Research Center of Metallurgical Engineering Technology of Shaanxi Province, Xi'an 710055, China
3 College of Materials Science and Engineering, Chongqing University, Chongqing 400044, China
4 Hanzhong Iron and Steel Co., Ltd., Shaanxi Iron and Steel Group, Hanzhong 724200, China; 17693110936@163.com (B.Z.); i142561@163.com (W.B.)
* Correspondence: shan.ren@cqu.edu.cn (S.R.); xaxxd@xauat.edu.cn (X.X.)

**Abstract:** Low-temperature NH$_3$-SCR is an efficient technology for NO$_x$ removal from flue gas. The carbon-based catalyst designed by using porous carbon material with great specific surface area and interconnected pores as the support to load the active components shows excellent NH$_3$-SCR performance and has a broad application prospect. However, overcoming the poor resistance of H$_2$O and SO$_2$ poisoning for carbon-based catalysts remains a great challenge. Notably, reviews on the sulfur and water resistance of carbon-based low-temperature NH$_3$-SCR catalysts have not been previously reported to the best of our knowledge. This review introduces the reaction mechanism of the NH$_3$-SCR process and the poisoning mechanism of SO$_2$ and H$_2$O to carbon-based catalysts. Strategies to improve the SO$_2$ and H$_2$O resistance of carbon-based catalysts in recent years are summarized through the effect of support, modification, structure control, preparation methods and reaction conditions. Perspective for the further development of carbon-based catalysts in NO$_x$ low-temperature SCR is proposed. This study provides a new insight and guidance into the design of low-temperature SCR catalysts resistant to SO$_2$ and H$_2$O in the future.

**Keywords:** low-temperature SCR; carbon-based catalysts; SO$_2$ and H$_2$O resistance; NO$_x$

## 1. Introduction

Nitrogen oxide (NO$_x$) mainly comes from iron and steel enterprises, electric power enterprises, and other fixed sources [1]. As one of the air pollutants, it not only reacts with hydrocarbons to form photochemical smog through ultraviolet irradiation, but it also reacts with water in the air to generate the main component of acid rain, causing a series of serious environmental problems [2–4]. This poses a serious threat to the human respiratory system. Therefore, it is an increasingly urgent task to remove NO$_x$. Currently, several denitration technologies, including selective non-catalytic reduction (SNCR) [5–7], selective catalytic reduction (SCR) [8,9], and non-selective catalytic reduction (NSCR) [10,11], can be used to reduce NO$_x$. Among them, selective catalytic reduction (SCR) for the removal of NO$_x$ is considered to be one of the most promising technologies which has been commercialized for both stationary and mobile NO$_x$ sources. The choice of catalyst is crucial for the application of SCR denitration technology. Depending on the efficiency of NO$_x$ removal, primarily the operating temperatures, process parameters, and catalyst regeneration, various catalysts including metal (titanium, vanadium, and iron, etc.) oxides [12], noble metals [13], zeolites [14], and activated carbon [15,16] have been developed for the NH$_3$-SCR process. At present, V$_2$O$_5$/TiO$_2$ [17,18] and V$_2$O$_5$-WO$_3$/TiO$_2$ [19] catalysts with high activity and sulfur resistance have been widely used in industrial denitration [20,21]. However, the traditional V-based catalyst temperature is 300–400 °C [22], which does not apply for most industrial flue gas cleaning with lower practical temperature. For example, in the iron

and steel industry, the temperature of the sintering flue gas is generally 120–180 °C. It needs to be heated to the application temperature to satisfy the high temperature catalyst, a process that consumes a lot of energy. The supports, such as $TiO_2$, $Al_2O_3$, $ZrO_2$, $SiO_2$, and carbon-based materials [23–25], have important influence on the denitration performance. Among them, carbon-based materials have gradually become a promising support for low-temperature SCR catalysts due to their large specific surface area, developed pore structure, and broad resources [26]. Compared with titanium, iron, aluminum, and other metal oxide catalysts, the surface of carbon-based catalysts is abundant in oxygen functional groups that can well adsorb $NO_x$. In addition, the larger specific surface area can provide a place for the active metal to be highly dispersed. The catalysts with rare earth oxides and transition metal oxides as active components have excellent catalytic performance at low temperature, such as $MnO_x/ACs$ [23,27], $CeO_x/ACs$ [28,29], $FeO_x/ACs$ [30], etc. In the flue gas, these carbon-based catalysts display high low-temperature SCR activity in atmospheres free of $SO_2$ and $H_2O$. In particular, the valence state of Mn-based catalysts is variable and the redox ability is superb, which display good denitration ability. However, when the flue gas contains $SO_2$ and $H_2O$, the denitration activity of catalysts decreases greatly [31–33]. For example, the actual sintering flue gas contains about 10–13% water vapor and 300–1500 mg/Nm$^3$ $SO_2$ [34]. Under the conditions, the catalyst is prone to inactivate, which limits the practical application [35,36]. Therefore, it is necessary to develop the carbon-based catalysts with high $SO_2$ and $H_2O$ resistance for low-temperature SCR technology.

Over time, the related scholars have conducted a lot of work on the sulfur and water resistance of low-temperature $NH_3$-SCR catalysts and achieved remarkable results. In addition, the reviews in this area have been widely reported. A review by Zhang et al. comprehensively overviewed the progress of Mn-based catalysts in sulfur and water resistance and focused on analyzing the challenges and opportunities faced in the development of Mn-based catalysts [37]. A review by Xu et al. provided the reaction mechanism of Ce-based catalysts for low-temperature $NH_3$-SCR, and the technology to improve the resistance to sulfur and water was emphasized [38]. The recent review by Tang et al. also summarized the research progress of Mn-based catalysts in improving the denitrification activity of $NH_3$-SCR at low temperature and the resistance to sulfur and water, and the challenges and possible solutions for designing catalyst systems with high sulfur and water resistance were discussed in detail [39]. However, a review of sulfur and water resistance performance of carbon-based catalysts for low-temperature $NH_3$-SCR is rarely reported. This paper has reviewed the research findings on carbon-based catalysts in sulfur and water resistance in recent years. The reaction behavior of the low-temperature SCR catalyst and the poisoning mechanism in the flue gas containing $SO_2$ and $H_2O$ are summarized. For carbon-based catalysts, strategies to enhance the $SO_2$ and $H_2O$ resistance performance are systematically introduced from the aspects of catalyst support, modification, structure control, and preparation methods. In addition, the application of theoretical calculation in the development of low-temperature SCR carbon-based catalysts resistant to $SO_2$ and $H_2O$ is introduced. Finally, the possible development direction in this field is proposed, which plays a certain reference and guidance role for the design of low-temperature carbon-based catalysts resistant to $SO_2$ and $H_2O$ in the future.

## 2. Reaction Mechanisms of Carbon-Based Catalysts for SCR Denitration

### 2.1. Reaction Mechanisms of Carbon-Based Catalysts

NO oxidation on carbon-based catalysts is a micropore filling process with NO as the adsorbate. $NH_3$-SCR mainly uses $NH_3$ as the reducing agent, which selectively reduces $NO_x$ to $N_2$ and $H_2O$ at a temperature of 80–400 °C. The catalytic process lowers the activation energy of the chemical reactions [36]. Typical $NH_3$-SCR reaction equations are shown in Equations (1) and (2).

$$4NO + 4NH_3 + O_2 \rightarrow 4N_2 + 6H_2O \tag{1}$$

$$2NO_2 + 4NH_3 + O_2 \rightarrow 3N_2 + 6H_2O \tag{2}$$

If the $O_2$ is absent, the reaction equations are shown as Equations (3)–(5).

$$6NO + 4NH_3 \rightarrow 5N_2 + 6H_2O \tag{3}$$

$$6NO_2 + 8NH_3 \rightarrow 7N_2 + 12H_2O \tag{4}$$

$$NO + NO_2 + 2NH_3 \rightarrow 2N_2 + 3H_2O \tag{5}$$

At present, there are two generally recognized mechanisms of the $NH_3$-SCR reaction, Eley–Rideal (E-R) [40] and Langmuir–Hinshelwood (L-H) [41,42] mechanisms.

For the E-R mechanism, $NH_3$-SCR of $NO_x$ occurs through Equations (6)–(8). $NH_3$ is adsorbed and oxidized to $NH_2$ by acid sites. Then, $NH_2$ active intermediates react with the gas phase NO to form $N_2$ and $H_2O$.

$$NH_3(g) \rightarrow NH_3(a) \tag{6}$$

$$NH_3(a) + O(a) \rightarrow NH_2(a) + OH(a) \tag{7}$$

$$NO(g) + NH_2(a) \rightarrow NH_2NO(a) \rightarrow N_2(g) + H_2O \tag{8}$$

The $NH_3$-SCR of $NO_x$ according to the L-H mechanism occurs via Equations (6), (9) and (10). NO reacts with $O_2$ to generate adsorbed $NO_2$, which reacts with coordinating $NH_3$ to form $N_2$ and $H_2O$.

$$NO + O_2(g) \rightarrow NO_2(a) \tag{9}$$

$$NO_2(a) + 2NH_3(a) + NO(g) \rightarrow 2N_2 + 3H_2O \tag{10}$$

Carbon-based materials show larger specific surface area and abundant pore structure, which provide more active sites for the reaction. More gas phases NO and $O_2$ are captured on the empty active sites. NO adsorbed on the carbon surface is oxidized to adsorbed $NO_2$. The adsorbed $NO_2$ further reacts with $NH_3$ and NO to generate $N_2$ and releases the active vacancies so that the whole reaction continues. Among them, the adsorption of $NO_2$ on the AC surface involves the participation of NO, and the adsorbed $NO_2$ oxidizes the AC surface, thereby promoting the subsequent $NO_2$ adsorption.

For carbon-based catalysts, at a temperature of 80–200 °C, the $NH_3$-SCR reaction mainly follows the L-H mechanism, or the two mechanisms cooperate. At a higher temperature above 200 °C, it mainly follows the E-R mechanism.

### 2.2. Poisoning Mechanisms of Carbon-Based Catalysts

The active temperature range of carbon-based $NH_3$-SCR catalysts is rather low, about 100–150 °C, which avoids the clogging of pores on the surface of the catalyst due to excessive temperature. However, they are easily poisoned by $H_2O$ and $SO_2$, leading to catalyst deactivation, which seriously affects their application and development in industry. Therefore, in order to improve $H_2O$ and $SO_2$ tolerance, relevant scholars have conducted a lot of research on the mechanism of $SO_2$ and $H_2O$ poisoning in the process of carbon-based catalyst denitration.

### 2.2.1. Poisoning of $SO_2$

The catalytic performance of carbon-based catalysts is severely affected by $SO_2$ [43–46]. The effects of $SO_2$ poisoning can be divided into reversible inactivation and irreversible

inactivation. The competitive adsorption of $SO_2$ and NO on carbon-based catalysts is a reversible deactivation. In an irreversible situation, when the adsorbed $SO_2$ reacts with the active components on the carbon-based catalysts' surface, it generates sulfate, which occupies the active center, resulting in a decrease in available active sites and $NO_x$ conversion (Figure 1). For $CeO_2$-$WO_3$/$TiO_2$ catalysts shown in Figure 2, the introduction of $SO_2$ prevents the generation of active intermediates, such as $NH_2NO$. In addition, $SO_2$ reacts with $NH_3$ or $CeO_2$ to form $(NH_4)_2SO_4$ and $Ce_2(SO_4)_3$, respectively, which cover the active sites and inhibit the redox performance. This results in the irreversible inactivation. In addition, industrial flue gas usually contains other metals such as Cd, and $SO_2$ binding with it also produces $CdSO_4$ to cover the active sites, causing further deactivation of the catalyst. The production of sulfate on carbon-based catalysts' surface covering the active sites and the vulcanization of the active metal are the two main forms of permanent deactivation. As shown in Figure 3, the addition of $SO_2$ to a $MnFeO_x$ catalyst results in the production of $MnSO_4$, $FeSO_4$, and $(NH_4)_2SO_4$, which inhibit both L-H and E-R mechanisms. Thus, the catalysts display poor SCR activity. When Sm is dropped, $SO_2$ is preferred to combine with it. As a result, the active sites could be protected. The addition of another metal that preferentially reacts with $SO_2$ is the main method to enhance the $SO_2$ resistance of carbon-based $NH_3$-SCR catalysts.

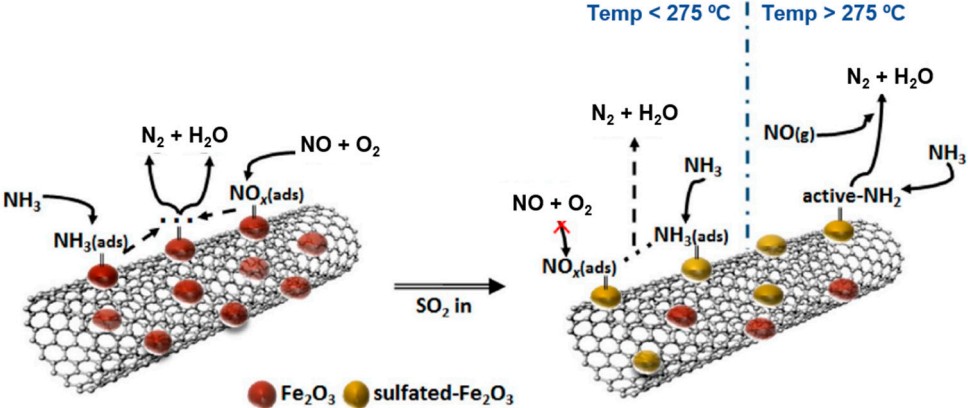

**Figure 1.** Reaction routes for fresh Fe/CNTs and $SO_2$-Fe/CNTs catalysts. (Reproduced with permission from reference [43], Copyright 2018, MDPI (Basel, Switzerland)).

### 2.2.2. Poisoning of $H_2O$

Water vapor deprives the catalyst surface of available active sites, thereby negatively affecting the SCR reaction [43,47]. Generally, the poisoning mechanism of $H_2O$ includes both reversible inactivation and irreversible inactivation. Among them, the competitive adsorption between $H_2O$ and $NH_3$ (or NO) is the main reason for the reversible deactivation (Figure 4). When the adsorption of reactive gas molecules is lower, the $NO_x$ conversion rate also decreases. At a lower temperature, the thermal stability of hydroxyl groups is poor, and water has a significant effect on the catalyst activity, resulting in irreversible deactivation. However, when the temperature is higher than 200 °C, water vapor is not easily adsorbed on the catalyst surface, and the effect of water on the catalyst activity can be ignored. Water vapor affects the crystal shape, grain size, and specific surface area of carbon-based catalysts to a certain extent, which results in a reduction in catalytic activity. The adsorption of water molecules on a carbon-based catalyst surface competes with reaction gases for the active sites, reducing the active centers available and thus decreasing the activity of the catalyst.

### 2.2.3. Poisoning of $SO_2$ and $H_2O$

So far, the poisoning mechanism of $H_2O$ and $SO_2$ of low-temperature SCR catalysts has been widely recognized, and the study of the mechanism of carbon-based catalysts resisting $SO_2$ and $H_2O$ has made some progress [48,49]. It needs to be emphasized that some catalysts have good resistance to single $SO_2$ or $H_2O$, and the denitration rate could be

maintained at a relatively high level. However, when $SO_2$ and $H_2O$ coexist, the denitration rate decreases sharply. The main reason is that the reaction of $H_2O$ and $SO_2$ produces $H_2SO_4$, which accelerates the sulfation of the active metal oxides. Therefore, we still need to conduct in-depth research on carbon-based catalysts with both sulfur and water resistance. As shown in Figure 5, when $SO_2$ is introduced, it is absorbed on the $MnO_x$ surface and oxidized to $SO_3$. When the sulphates accumulate to a certain amount, the formation of $SO_4^{2-}$ polymers result in a reduction in the surface area and the inhibition of redox property. Washing with water can remove $SO_4^{2-}$ polymers and restore the catalytic activity of active components.

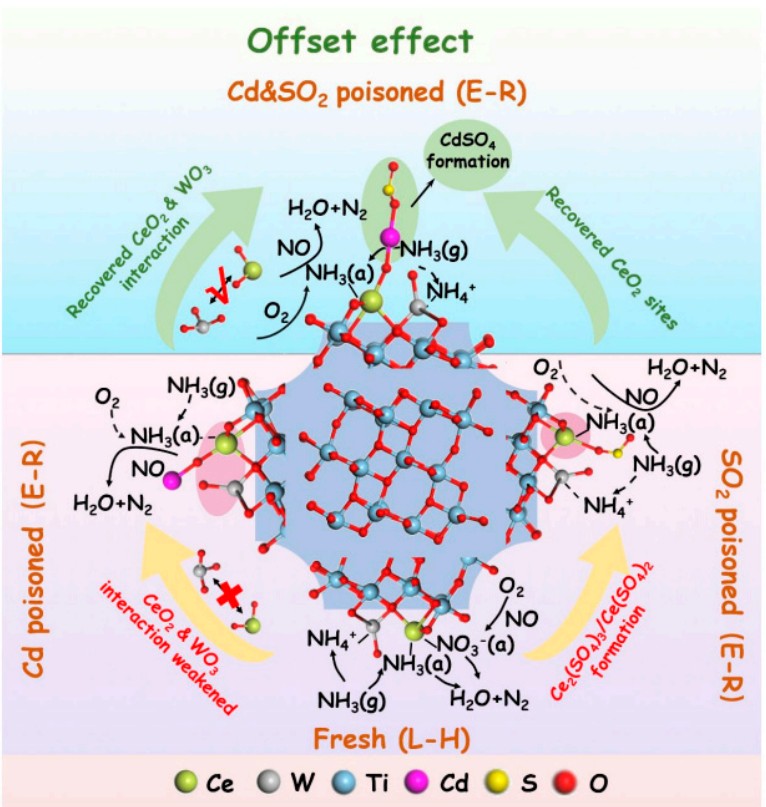

**Figure 2.** Offset effects between Cd and $SO_2$ over $CeO_2$–$WO_3$/$TiO_2$ catalysts. (Reproduced with permission from reference [44], Copyright 2020, American Chemical Society (Washington, DC, USA)).

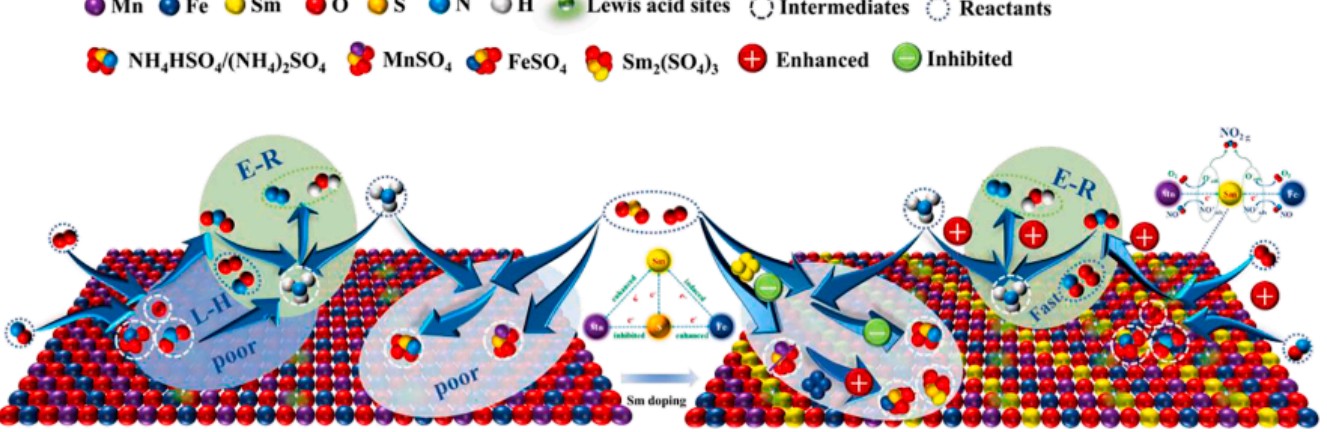

**Figure 3.** Mechanism model of Sm promoting SCR activity and $SO_2$ resistance over $MnFeO_x$ catalysts. (Reproduced with permission from reference [46], Copyright 2022, Elsevier (Amsterdam, The Netherlands)).

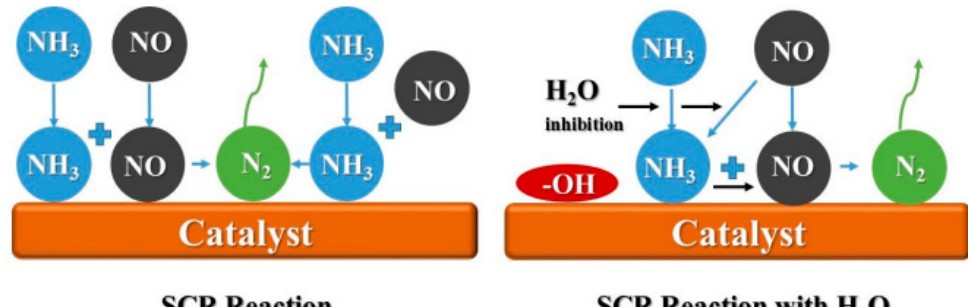

**Figure 4.** Scheme of regular SCR reaction and $H_2O$ poisoning effect. (Reproduced with permission from reference [43], Copyright 2018, MDPI (Basel, Switzerland)).

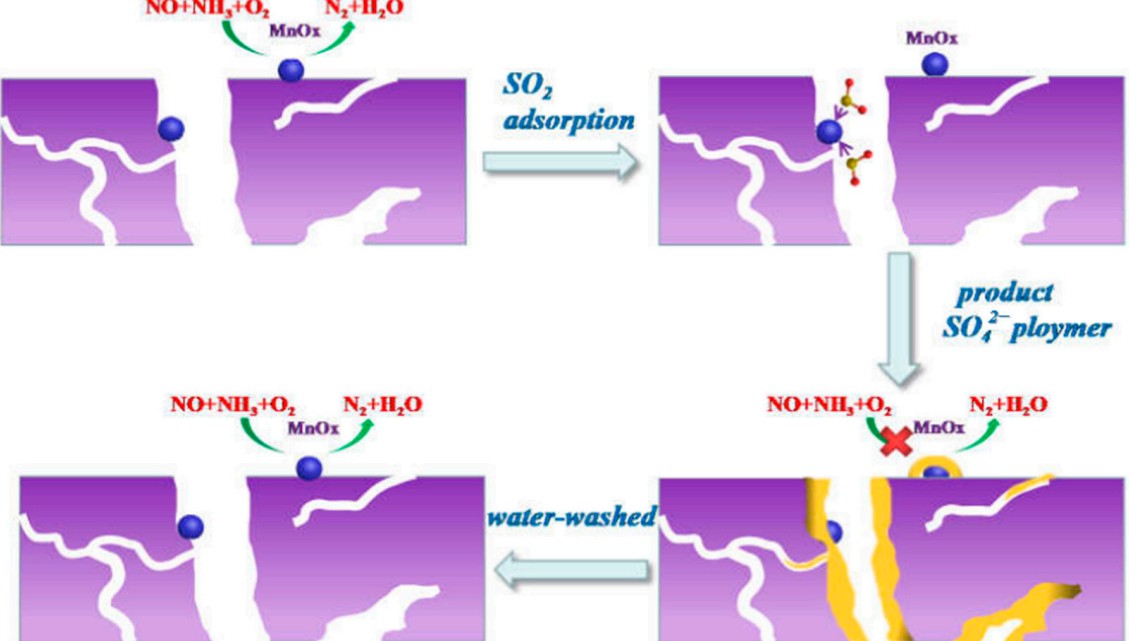

**Figure 5.** Proposed sulfur–treated poisoning and water–washed regeneration mechanism model. (Reproduced with permission from reference [48], Copyright 2021, MDPI (Basel, Switzerland)).

## 3. Research Progresses of Carbon-Based Catalysts for $SO_2$ and $H_2O$ Resistance

In order to improve the denitration activity and enhance the anti-toxic ability of the catalysts, the finding of suitable supports and active components is necessary [50]. The research progress of $SO_2$ and $H_2O$ resistant carbon-based catalysts was introduced as follows [51,52].

### 3.1. Effects of Support

In industrial applications, in terms of thermal stability, mechanical strength, and large specific surface area, the supported catalyst is superior to the non-supported oxide catalyst. Selecting a suitable support is an important way to improve the anti-poisoning ability of carbon-based catalysts. In the presence of $O_2$, $SO_2$ will react with $NH_3$ to form $(NH_4)_2SO_4$, leading to the blockage of the pores of the catalyst, and $H_2O$ is adsorbed on the surface of the support to inhibit the adsorption of reaction gas. It is noted that the number of acid centers affects the adsorption performance of $SO_2$. Therefore, an important method to enhance the resistance of $SO_2$ and $H_2O$ is to improve the surface acidity of the support [53–55]. At present, the commonly used carbon support mainly includes activated carbon, carbon nanotubes, activated carbon fiber, and graphene. The carbon-based catalysts for SCR described above are summarized in Table 1.

**Table 1.** The carbon-based catalysts for SCR.

| Catalysts | Active Test Reaction Conditions | NO$_x$ Conversion/ Temperature | N$_2$ Selectivity/ Temperature | SO$_2$ Tolerance/ Water-Resistance | Ref. |
|---|---|---|---|---|---|
| Activated carbon/coke | [NH$_3$] = 550 ppm<br>[NO] = 500 ppm<br>[O$_2$] = 5 vol%<br>[SO$_2$] = 50 ppm<br>[H$_2$O] = 10 vol%<br>balance of N$_2$<br>GHSV = 50,000 h$^{-1}$ | >40%/<br>(150 °C) | - | >38%/<br>>32% | [51] |
| Carbon nanotubes | [NH$_3$] = 500 ppm<br>[NO] = 500 ppm<br>[O$_2$] = 5 vol%<br>[H$_2$O] = 3 vol%<br>balance of N$_2$<br>GHSV = 60,000 h$^{-1}$ | >80%/<br>(200 °C) | >60% | - | [56] |
| Activated carbon fiber | [NH$_3$] = 500 ppm<br>[NO] = 500 ppm<br>[O$_2$] = 5 vol%<br>balance of N$_2$<br>GHSV = 40,000 h$^{-1}$ | >75%/<br>(150 °C) | - | - | [57] |
| Graphene | [NH$_3$] = 500 ppm<br>[NO] = 500 ppm<br>[O$_2$] = 7 vol%<br>[H$_2$O] = 10 vol%<br>balance of N$_2$<br>GHSV = 67,000 h$^{-1}$ | >90%/<br>(150 °C) | >90% | >80%/<br>>95% | [58] |

### 3.1.1. Activated Carbon/Coke

Due to the advantages of abundant surface groups, large specific surface area, high porosity, and strong adsorption capacity, activated carbon/coke (AC) has attracted extensive attention and is widely used as catalyst support [51]. As shown in Figure 6, the reaction of NO on MnO$_x$–Cu/AC catalysts follows both L-H and E-R mechanisms. For the E-R mechanism, NO in the gas phase reacts with NH$_3$ activated by the acidic sites on the catalyst surface. For the L-H mechanism, NO and O$_2$ in the gas phase interact and are oxidized to nitrate or nitrite intermediates by the catalyst, which then react with activated NH$_3$. It is worth noting that regardless of the mechanism, the activation of NH$_3$ at the acidic site is a key step for the reaction to proceed. The adsorption properties of AC are impacted by changing the polarity and/or acidity of surface functional groups. The main active sites on activated carbon are different kinds of oxygen-containing groups. Some researchers believe that for the V$_2$O$_5$/AC catalyst, in the presence of H$_2$O, SO$_2$ will react with NH$_3$ and O$_2$ to form ammonium sulfate, causing the pores of the catalyst to be blocked. As shown in Figure 7, after the introduction of SO$_2$, the activity of the V$_2$O$_5$/AC catalyst decreased rapidly, then had a weak rise, and finally declined slowly to remain a stable value [53]. However, when the SO$_2$-poisoned V$_2$O$_5$/AC catalyst was characterized, it is observed that only a small part of the pores on surface are blocked by sulfate. The generation of VOSO$_4$ via the reaction of V$_2$O$_5$ and SO$_2$ caused the NO conversion to drop sharply, which is the main reason for the deactivation of the catalyst. The sulfurization of active metal oxides by SO$_2$ is the main cause of the irreversible inactivation of most carbon-based catalysts in sulfur-containing flue gas. For Mn–Ce/AC catalysts shown in Figure 8, SO$_2$ reacted with NH$_3$ to inhibit the reaction of NO and NH$_3$ [54]. On the other hand, SO$_2$ combined with manganese oxide and cerium oxide to form MnSO$_4$ and (Ce)$_2$(SO$_4$)$_3$, which caused permanent inactivation. The addition of V could improve the acidity of the catalyst surface and inhibit the combination of SO$_2$ and NH$_3$. In addition, the vulcanization of Mn–Ce solid solution is also prevented. Therefore, the sulfur resistance of the catalyst could be enhanced.

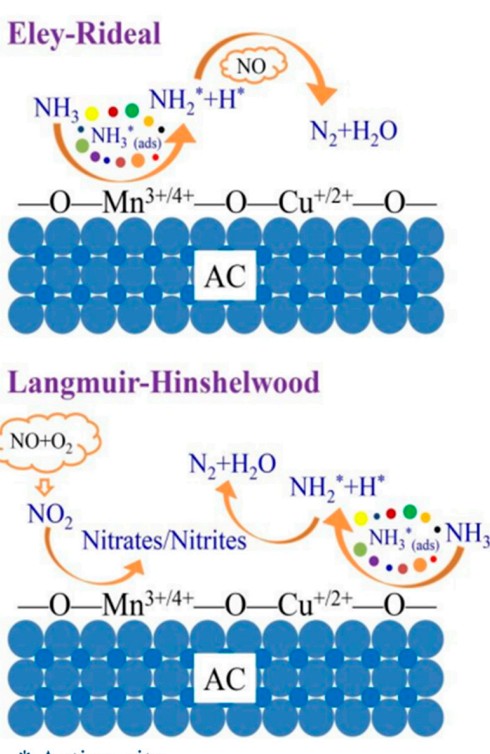

**Figure 6.** NO removal mechanism of the $MnO_x$–Cu/AC catalyst. (Reproduced with permission from reference [51], Copyright 2020, MDPI (Basel, Switzerland)).

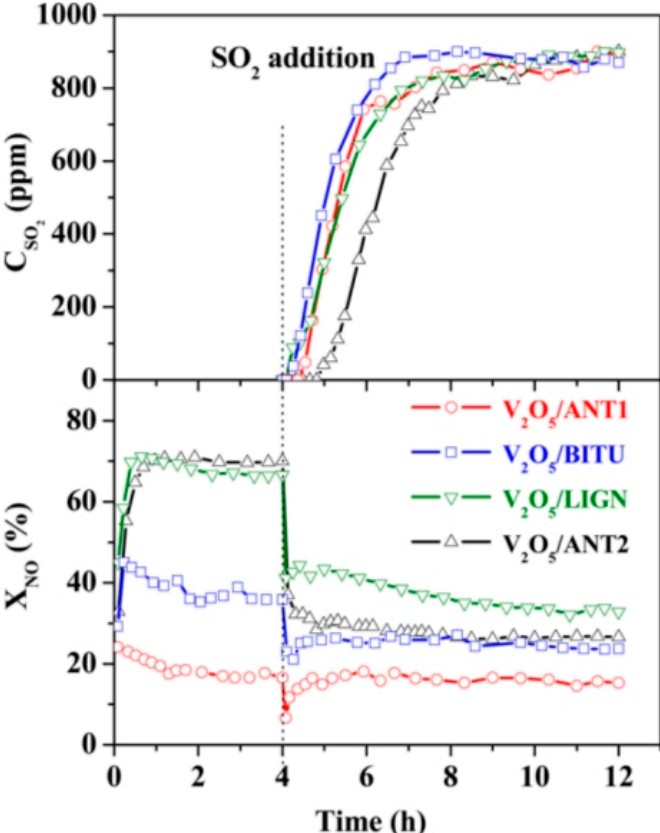

**Figure 7.** NO conversions and $SO_2$ release concentrations during the SCR of NO over various $V_2O_5$/AC catalysts at 200 °C. (Reproduced with permission from reference [53], Copyright 2014, American Chemical Society (Washington, DC, USA)).

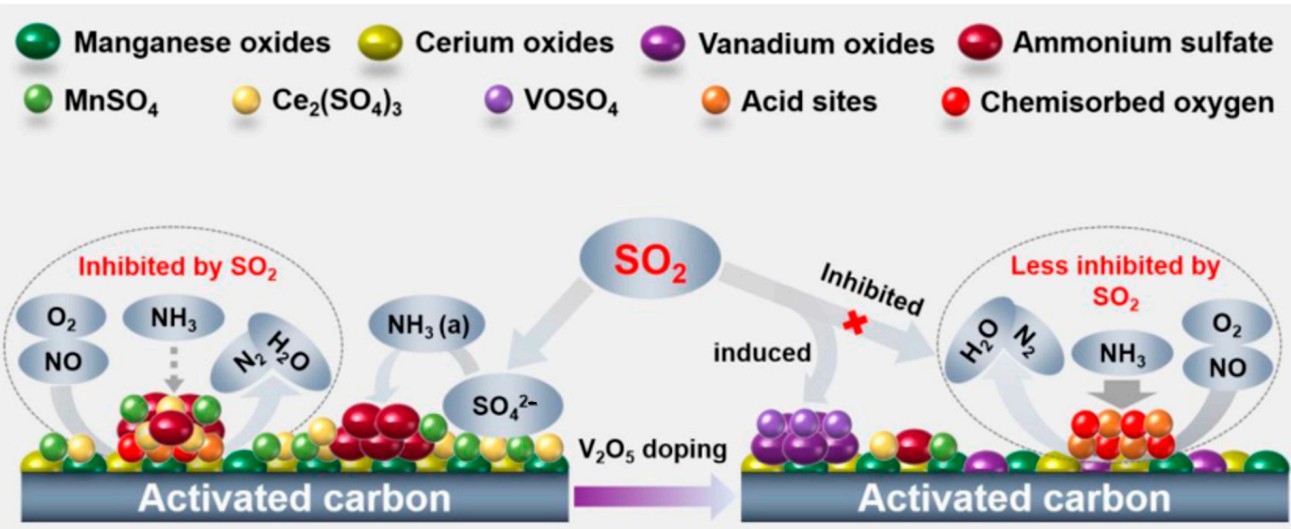

**Figure 8.** Mechanism of SO$_2$ tolerance over Mn–Ce(0.4)−V/AC catalysts. (Reproduced with permission from reference [54], Copyright 2019, Elsevier (Amsterdam, The Netherlands)).

In addition, different types of activated carbons have different specific surface areas and pore volumes. Among them, Mn–Ce catalysts supported by charcoal and coal exhibit a large specific surface area and strong surface acidity, which leads to an increase in adsorbed oxygen and high dispersibility of metal oxides with high catalytic activity. Meanwhile, the surface properties of AC have an obvious influence on the structure and performance of the supported catalysts. Nitric acid treatment can enhance the surface acidity of activated carbon and the dispersibility of active components and can significantly improve the NO conversion and SO$_2$ resistance of the catalyst [56]. However, AC catalysts are still prone to deactivation when SO$_2$ reaches a certain level. Therefore, further efforts should be made to strengthen the SO$_2$ resistance of SCR catalysts.

3.1.2. Carbon Nanotubes

As an allotrope of carbon, the pore size of carbon nanotubes (CNTs) can range from several nanometers to 100 nm. CNTs have adjustable nano hollow tubular structures, with a specific surface area of generally about 50–1300 m$^2$/g and a tensile strength of 50–200 GPa [59,60]. CNTs are used as supports for NH$_3$-SCR catalysts due to their high adsorption capacity of NO$_x$ and NH$_3$ as well as good SO$_2$ and H$_2$O resistance (Figure 9). At present, the research on CNT supports mainly focuses on the morphology of CNTs and the direction of CNTs composite supports. Wang et al. [61] elucidated the effects of SO$_2$ and water vapor on Ce/AC-CNTs catalysts. Obviously, when adding 50 ppm of SO$_2$, the Ce/AC-CNTs catalyst had a small inhibition on NO conversion, which was only reduced by 4.8%. In addition, when H$_2$O was introduced, the activity of Ce/AC-CNTs reduced by 21.7%, while the NO conversion of Ce/AC declined by 30.6%. CNTs can not only change the oxidation state of surface oxygen and active components but also increase the concentration of chemically adsorbed oxygen species. CNTs have a certain resistance to SO$_2$ and H$_2$O [62]. The morphology of CNTs is also related to the performance of the catalyst. When SO$_2$ and H$_2$O coexist, the denitration rate of multi-shell CNT catalysts is nearly 60% higher than that of ordinary CNT catalysts. Multi-shell CNTs can effectively inhibit the generation of surface sulfate species [63]. However, due to the limitations of the preparation process and scale of CNTs, the cost of CNTs is very high, which is known as "black gold". The use of CNTs as catalyst supports is still in the stage of laboratory research and has not been industrialized.

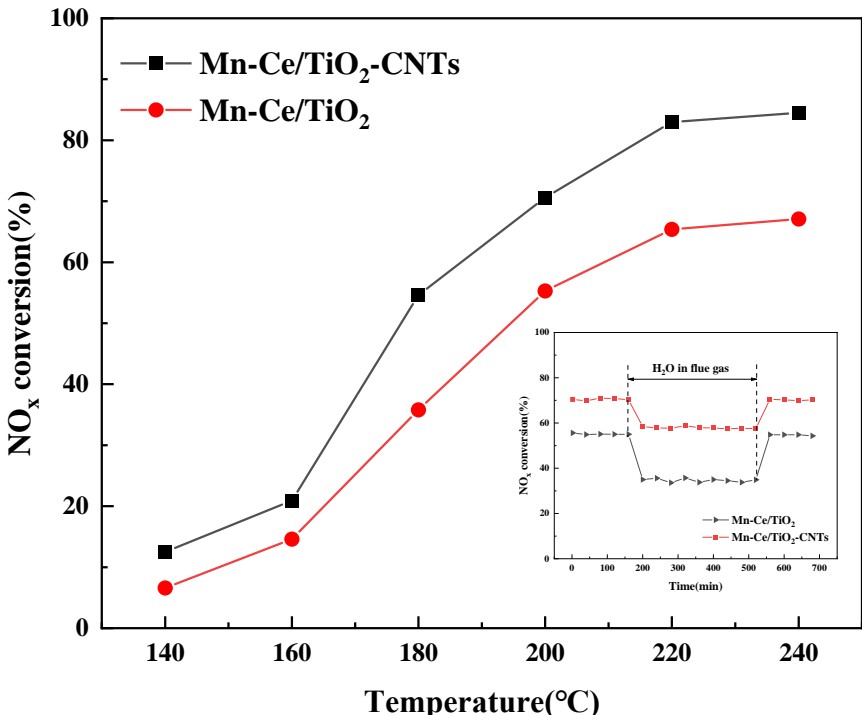

**Figure 9.** NO$_x$ conversion over Mn–Ce/TiO$_2$ and Mn–Ce/TiO$_2$–CNTs catalysts. (Reproduced with permission from reference [59], Copyright 2021, Elsevier (Amsterdam, The Netherlands)).

### 3.1.3. Activated Carbon Fiber

Activated carbon fiber (ACF) is usually a porous fiber material that is activated by carbon fiber or various carbon-containing materials at high temperature. ACF has a large specific surface area which can reach 1000–1500 m$^2$/g or even more than 2000 m$^2$/g. Among the pores of ACF, the proportion of micropores accounts for more than 90%, most of which are distributed on the fiber surface with narrow pore size, uniform distribution, easy contact with adsorbates, and strong adsorption capacity for low-concentration adsorbates. Hence, it has a fast adsorption and desorption rate and excellent adsorption and separation performance.

ACF has a certain catalytic activity and strong adsorption capacity for NO, and the adsorbed NO and O$_2$ in micropores can be converted into NO$_2$. However, the removal efficiency of NO using ACF as a catalyst alone is low. The highest NO conversion of ACF in the range of 150–310 °C is only about 15%. When 10% CeO$_2$ is added to ACF, the denitration activity is greatly improved. In the temperature range of 140–240 °C, more than 85% of the denitration rate can be achieved, which is higher than that of the same amount of the MnO$_x$ catalyst [64]. The catalytic activity of ACF can be greatly increased by modifying it with active components (Figure 10). However, the cost of catalysts supported by activated carbon fiber is also very high, and it has not been applied in industry at present.

### 3.1.4. Graphene

Compared with other carbonaceous supports, Graphene (GE), as a novel type of carbon nanomaterial, has a unique planar extension structure with a specific surface area of 100–2600 m$^2$/g and conductivity of 10$^6$–10$^8$ S/m [65]. The excellent electronic properties can promote electron transfer in the redox process, which accelerates the catalytic reaction. The special structure of GE can disperse active components to improve the interaction, which produces more active oxygen species and active sites to enhance the low-temperature catalytic activity. TiO$_2$-GE nanocomposite support shows uniform components, which is conducive to increasing the specific surface area of the carrier, changing its pore structure characteristics, and promoting the uniform loading of active ingredients. At 180 °C, the NO

conversion of the CeO$_x$–MnO$_x$/TiO$_2$-GE catalyst reaches 95% [66]. As a catalyst support, GE can also inhibit the sulfation of active components. Compared with the unsupported catalyst [67], the GE catalyst has a strong resistance ability for H$_2$O and SO$_2$ (Figure 11). The NO conversion is restored from 73% to 79% at 180 °C after the H$_2$O and SO$_2$ is stopped. However, the existing graphene preparation process is immature, which limits the large-scale application.

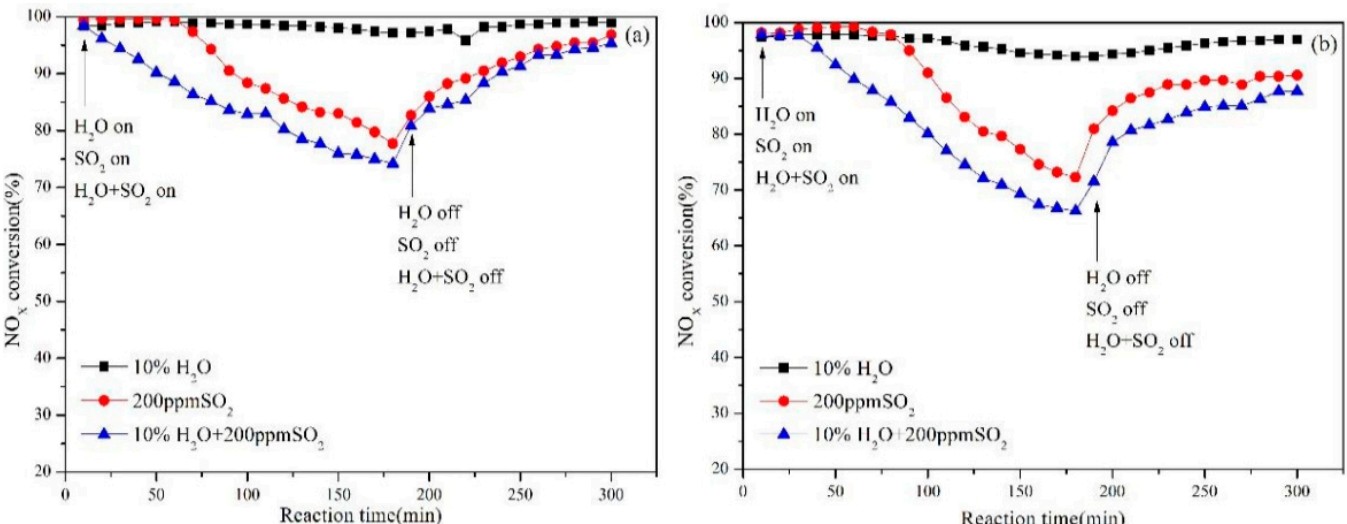

**Figure 10.** Effects of H$_2$O or/and SO$_2$ on NO$_x$ conversion of (**a**) MnO$_x$–CeO$_2$ (8:1)/GR and (**b**) MnO$_x$–CeO$_2$ (2:1)/GR at 140 °C. (Reproduced with permission from reference [64], Copyright 2017, Elsevier (Amsterdam, The Netherlands)).

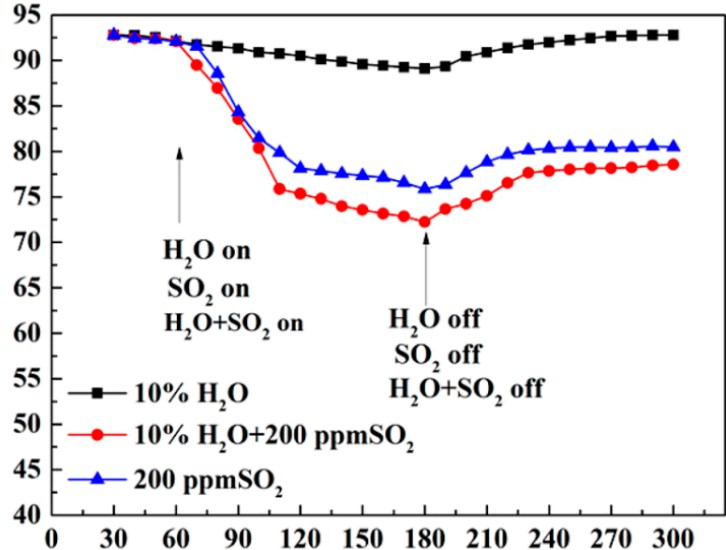

**Figure 11.** Effect of H$_2$O and SO$_2$ on NO$_x$ conversion over 7%MnO$_x$/TiO$_2$_0.8%GE catalysts. (Reproduced with permission from reference [67], Copyright 2014, American Chemical Society (Washington, DC, USA)).

### 3.2. Effects of Modification

Many scholars have studied the influence of active components on the anti-toxicity and denitration performance of the catalysts [68,69]. At present, more research is focused on Mn-based catalysts [70] and Ce-based catalysts [71]. Additionally, there are other metal oxide catalysts such as V and Fe catalysts. This section mainly introduces the related

research on the single modification, bimetallic modification, and polymetallic modification of metal oxides.

### 3.2.1. Single Metal Oxide-Modified Catalysts

Single metal-modified catalysts mainly include Mn-modified catalysts, Ce-modified catalysts, Fe-modified catalysts, V-modified catalysts, and other modified catalysts.

#### Single Mn-Modified Catalysts

Mn-based catalysts, as a research hotspot, have a variety of valence states and excellent redox ability, easily carry out the oxidation reaction, and have higher catalytic performance at low temperatures.

Generally, the higher valence state shows the better catalytic effect for Mn-based catalysts. The SCR activity decreases as follows: $MnO_2 > Mn_5O_8 > Mn_2O_3 > Mn_3O_4$ [72]. $MnO_x$-modified catalysts can increase the dispersity of amorphous states, the ratio of $Mn^{4+}/Mn^{3+}$, and the surface area and pore volume, which improves the NO conversion [73]. The performance of the catalyst is enhanced by adding Mn in an amount in the range of 10–20 wt.%. When the content of Mn increases to 25 wt.%, $MnO_x$ particles gather on the catalyst surface and the NO conversion decreases (Figure 12). However, $SO_2$ has a serious toxicity to single Mn-based catalysts. The active Mn atoms are sulfated after the introduction of $SO_2$, which leads to rapid deactivation. It is usually difficult to regenerate for the deactivated Mn-based catalysts.

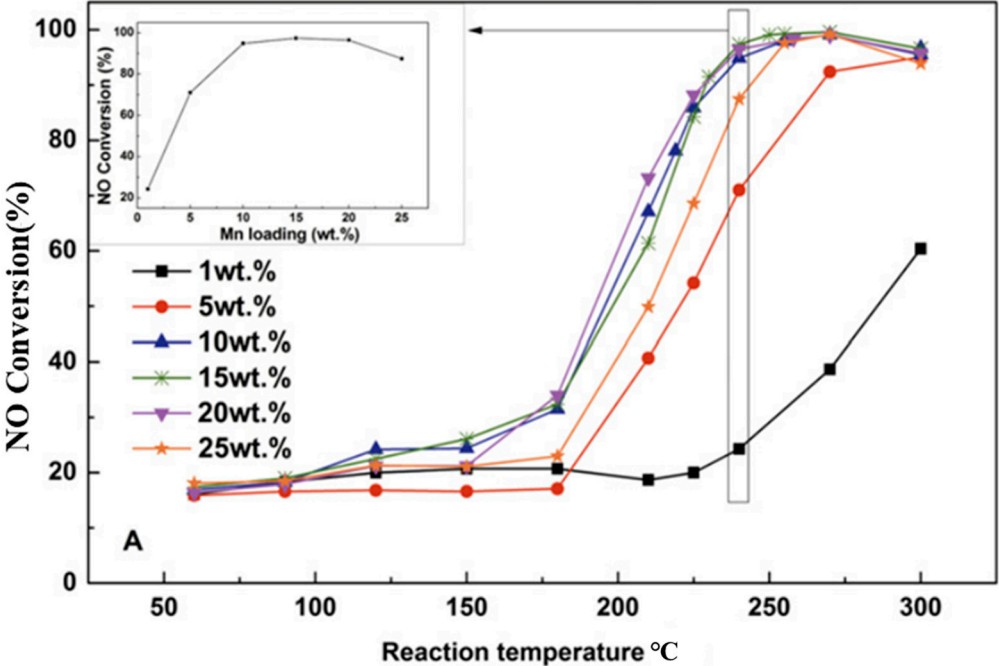

**Figure 12.** Effect of Mn loading of $MnO_x$/MWCNTs (60–100, 500) catalysts on catalytic activity. (Reproduced with permission from reference [68], Copyright 2021, Elsevier (Amsterdam, The Netherlands)).

#### Single Ce-Modified Catalysts

As an important rare earth metal oxide, $CeO_2$ has strong redox performance and excellent oxygen storage/release ability. Between $Ce^{4+}$ and $Ce^{3+}$, the electron transfer promotes an increase in active oxygen species and accelerates the conversion of NO to $NO_2$, thereby improving the activity of SCR. Therefore, $CeO_2$ is usually used as an active component to improve $NH_3$-SCR activity. Doping the rare earth element Ce effectively facilitates the generation of oxygen vacancies. The high concentration of oxygen vacancy is favorable for the adsorption of $O_2$ and the further oxidation of NO, which is conducive to the subsequent

reduction reaction [38]. In all temperature ranges, the maximum NO conversion of $CeO_2$ nanoparticles is only 50%. The NO conversion on pure carbon nanotubes is extremely low. While carbon nanotubes are introduced as supports for $CeO_2$, the NO conversion rate is increased significantly for all catalysts. In the temperature range of 250 °C to 370 °C, the $CeO_2$/CNTs platinum catalysts display the best activity, corresponding to above 90% NO conversion. For $CeO_2$/CNTs-PM catalysts, the maximum conversion of NO can only reach 80% below 380 °C [74]. The catalysts with a lower addition have less dispersed active sites, an incomplete catalytic reduction reaction, and low denitration efficiency. Appropriate $CeO_2$ content can enhance the active sites on the catalyst surface and improve the catalytic performance. Excessive $CeO_2$ loading causes the $CeO_2$ particles to aggregate on the surface, which leads to a reduction in active sites. Compared with $CeO_2$ nanoparticles, $CeO_2$ nanotubes on the catalyst surface have more Ce and O atoms, more acidic centers, and stronger acidity, which is conducive to improve SCR performance [75]. The $SO_2$ can preferentially bind with $CeO_2$ to form $Ce_2(SO_4)_3$, thereby reducing the generation of $(NH_4)_2SO_4$ and $NH_4HSO_4$ to suppress catalyst deactivation (Figure 13) [76]. Therefore, $CeO_2$-modified catalysts have broad prospects.

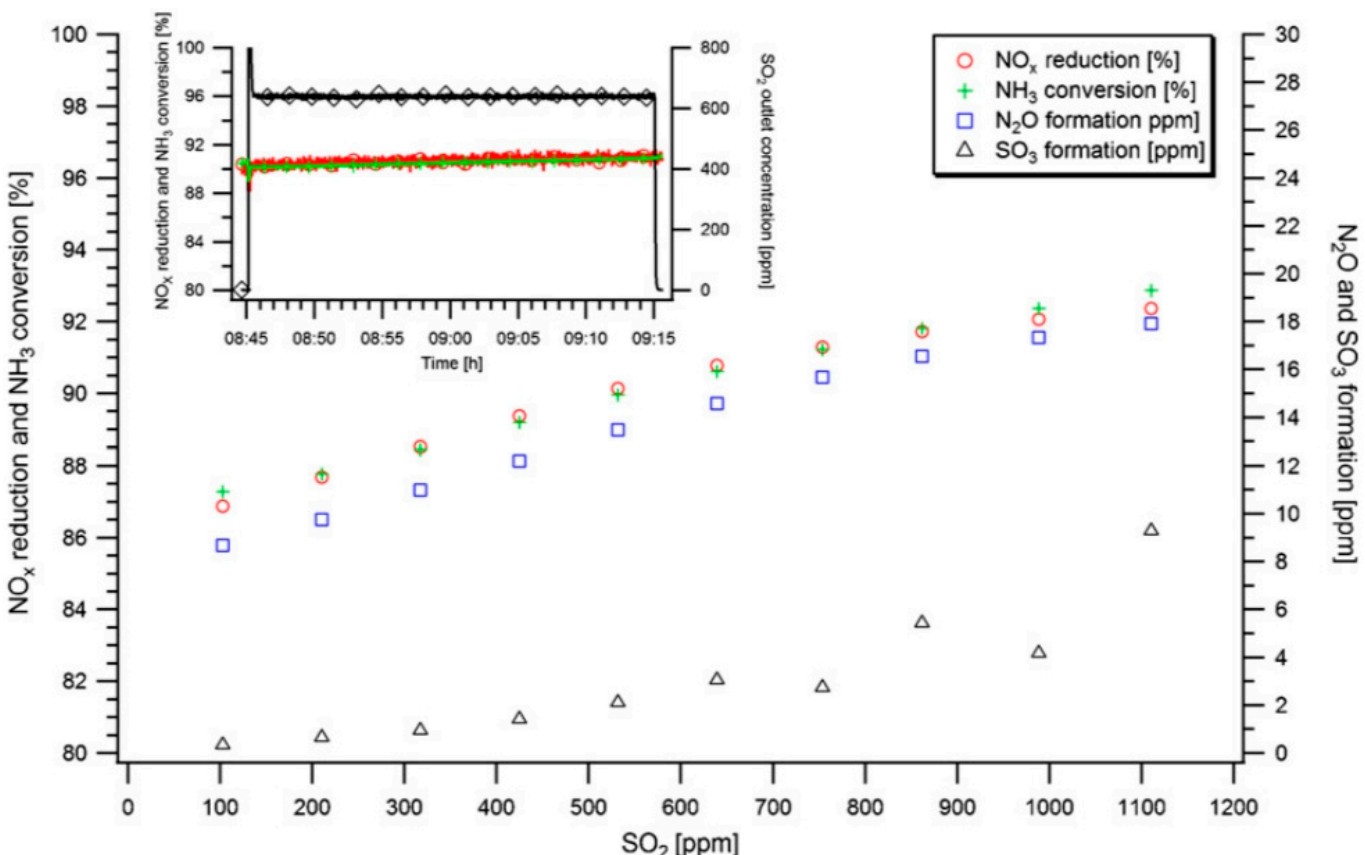

**Figure 13.** Influence of the $SO_2$ concentration on the $NO_x$ reduction performance over sample of $V_2O_5$-$WO_3$/$TiO_2$. (Reproduced with permission from reference [76], Copyright 2012, Elsevier (Amsterdam, The Netherlands)).

Single Fe-Modified Catalysts

Iron in $Fe_2O_3$ has a variety of valence states such as +2 and +3, which shows good redox properties. In addition, it has the characteristics of non-toxic and harmless, wide source, and low price. The mutual conversion between $Fe^{2+}$ and $Fe^{3+}$ can form unstable oxygen vacancies and lattice oxygen species with high transfer ability. Therefore, Fe is often used as the active component and support of catalysts and has attracted extensive attention among scholars [77–79]. For Fe-modified AC catalysts, when the molar ratio of

Fe to AC was 0.10 [80], the performance was optimal, exhibiting 83.9% NO conversion at 240 °C. However, when the ratio was 0.15, the $Fe_2O_3$/AC catalyst displayed poor NO conversion. The influences of $SO_2$, $H_2O$, or $SO_2 + H_2O$ on the $NO_x$ conversions of 8Mn6Fe/AC catalysts were investigated in [81]. The $NO_x$ conversion decreased by 12% when $H_2O$ was introduced into the reactor, due to a competitive adsorption between $NH_3$ and $H_2O$, and then, the activity remained stable. Fe was mainly dispersed in the form of $\gamma$-$Fe_2O_3$. After the introduction of Mn, the ratio of $Fe^{3+}$/$Fe^{2+} + Fe^{3+}$ changed only a little, and the value of $O_\beta$/($O_\beta + O_\alpha$) increased significantly. The doping of Mn increased the amount of chemisorbed oxygen with higher migration ability, which could oxidize NO to $NO_2$ to form a "fast SCR" reaction, thus promoting the denitration performance and $SO_2$ resistance of the catalysts (Figure 14).

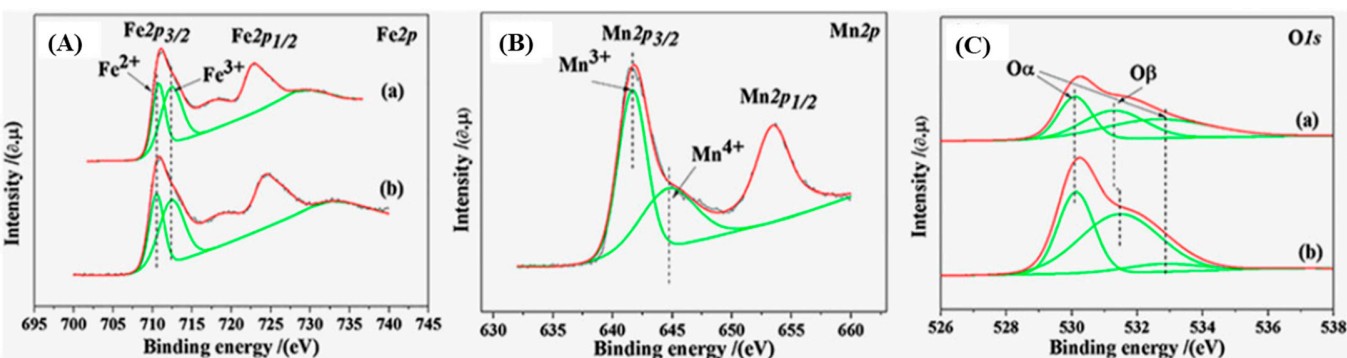

**Figure 14.** XPS spectra for (**A**) Fe 2p, (**B**) Mn 2p, and (**C**) O 1 s of $Fe_2O_3$/AC (**a**) and Mn–$Fe_2O_3$/AC (**b**) catalysts. (Reproduced with permission from reference [81], Copyright 2018, Wiley-Blackwell (Hoboken, NJ, USA)).

Single V-Modified Catalysts

Vanadium is a rich material with low cost and high energy efficiency which can be loaded on the support to increase the activity and anti-poisoning ability of catalysts. In the field of $NH_3$-SCR [82,83], traditional vanadium-titanium catalysts are widely applied, which makes more researchers explore vanadium-titanium catalysts that meet the anti-poisoning performance and low-temperature denitration performance.

$V_2O_5$/AC catalysts have a high $SO_x$ absorption capacity at 120–200 °C [84]. When the V addition is 5%, the NO conversion can reach 80% at 250 °C. After adding $SO_2$, the NO conversion increases to more than 90%. For the abnormal phenomenon of increasing NO conversion, it is found that the interaction of AC with vanadium oxide activates the ammonium sulfate on the low-loaded $V_2O_5$/AC catalyst. The activated species react with NO, increase the acidity on the surface, and enhance the catalytic activity. Catalysts with $V_2O_5$ as the active component do not perform as well as Mn-based catalysts in low-temperature SCR reactions, but its sulfur resistance is obviously higher than Mn-based catalysts.

Other Single-Modified Catalysts

In addition to $MnO_x$, $CeO_2$, $FeO_x$, and $V_2O_5$ modification, there are also some other modified catalysts which can also improve the denitration performance and poisoning resistance [85–87]. For nickel-supported carbon-based catalysts, appropriate pore structure and surface area can improve the $SO_2$ resistance. Doping the transition metal oxide $CrO_3$ on activated carbon can increase the formation of acid sites. Due to the valence change between $Cr^{6+}$ and its low oxidation states, such as $Cr^{5+}$, $Cr^{3+}$, and $Cr^{2+}$, the reaction rate of SCR is increased. When the mass ratio of Cr/AC is 2%, the NO removal efficiency is optimal, and the NO conversion rate is more than 90% at 125–150 °C. The NO conversion on the catalyst decreases after the addition of 100 ppm $SO_2$. When 300 ppm $SO_2$ is added, the NO conversion reduces to only about 5%. When 5% $H_2O$ is added, the NO conversion

decreases by about 15%. The NO conversion decreases to 71.5% within a few minutes under the conditions of 100 ppm $SO_2$ and 5% $H_2O$ [88]. The coexistence of $H_2O$ and $SO_2$ leads to the formation and deposition of ammonium sulfate, which blocks active sites on the catalysts and inactivates the catalysts.

However, single-modified catalysts show weak surface acidity, relatively low activity, a narrow working temperature window, and poor resistance of $SO_2$ under high temperature, which limit their practical application.

### 3.2.2. Bimetallic-Modified Catalysts

Bimetallic-modified catalysts mainly introduce Mn–Ce catalysts, Fe–Mn catalysts, V–Mn catalysts, and other bimetallic-modified catalysts.

### Mn–Ce Catalyst

Manganese-containing catalysts show excellent catalytic activity, while $SO_2$ resistance ability is poor. $CeO_2$ has excellent redox properties and certain sulfur resistance. The addition of Ce can enhance the oxygen storage capacity of the catalyst and the capacity of NO oxidation to $NO_2$. The NO conversion of $MnO_x$–$CeO_2$/AC catalysts is maintained at 92% at 210 °C after adding $SO_2$, which has excellent resistance to $SO_2$ toxicity [89]. The sulfation of active components is not obvious within a short time at 210 °C, and there is no obvious accumulation of $NH_4HSO_4$ in the reaction process. $CeO_2$ could reduce the bond energy of $NH_4^+$ and $SO_4^{2-}$ and lower the decomposition temperature of $NH_4HSO_4$ to enhance the $SO_2$ resistance. When the support is carbon nanotubes, the Mn–CeCNTs-R catalyst has good dispersibility of active components on its surface [90]. Under the conditions of 4 vol% $H_2O$ and 100 ppm $SO_2$, it shows strong tolerance. With the coexistence of $H_2O$ and $SO_2$, the NO conversion of the catalyst decreases by 13%. The NO conversion rate returns to 90% after stopping the addition of $SO_2$ and $H_2O$ (Figure 15). Compared with the single metal-modified catalysts, the resistance to $SO_2$ of Mn–Ce bimetallic catalysts is significantly improved.

### Fe–Mn Catalyst

At low temperatures, Fe–Mn bimetallic oxides have high NO conversion and $SO_2$ resistance. The NO conversion of Mn–$FeO_x$/CNTs catalysts is close to 98% at 180 °C and decreases by 20% after adding $SO_2$ [91]. By being in the form of amorphous oxides, Fe and Mn are highly dispersed on the support surface so that the catalyst has good redox performance. The NO conversion of $Fe_2O_3$@$MnO_x$@CNTs catalysts decreases from 95% to 91% and recovers to 95% after $SO_2$ and $H_2O$ removal, while $MnO_x$@CNTs catalysts only recovered to 36% and could not be restored to the initial level [60]. The resistance of Fe–Mn bimetallic catalysts to $SO_2$ and $H_2O$ is better than Mn–Ce bimetallic catalysts.

### Other Bimetallic-Modified Catalysts

In addition to the above bimetallic catalysts, other bimetallic catalysts can also enhance resistance to $H_2O$ and $SO_2$ performance. For instance, CeMo(0.3) hollow microsphere catalysts prepared with carbon microspheres as a template not only have the best low-temperature SCR performance [92] but have good stability and $H_2O$ resistance as well (Figure 16). For $Fe_2O_3$/AC catalysts, doping Ce can make $\gamma$-$Fe_2O_3$ evenly disperse on the AC surface, and the denitrification efficiency is significantly improved. Moreover, the sulfur resistance can be enhanced. The catalysts with a Ce/Fe mass ratio of 0.5:6 can achieve 94.1% NO conversion when $100 \times 10^{-6}$ (vol) $SO_2$ is added at 240 °C. When $H_2O$ is 5 wt.%, the NO conversion is stable at 86%. The introducing of slight vanadium oxide can enhance the dispersity of Fe species on $Fe_2O_3$/AC catalysts [93]. Whether it is V-modified or non-V-modified, in the presence of $H_2O$ and $SO_2$, catalysts are severely deactivated. But at low space velocities, the inhibition caused by $H_2O$ and $SO_2$ is reversible for 3%Fe-0.5%V catalysts. The addition of V species prevents the generation of sulfate by increasing the surface acidity, thereby enhancing the resistance to $SO_2$.

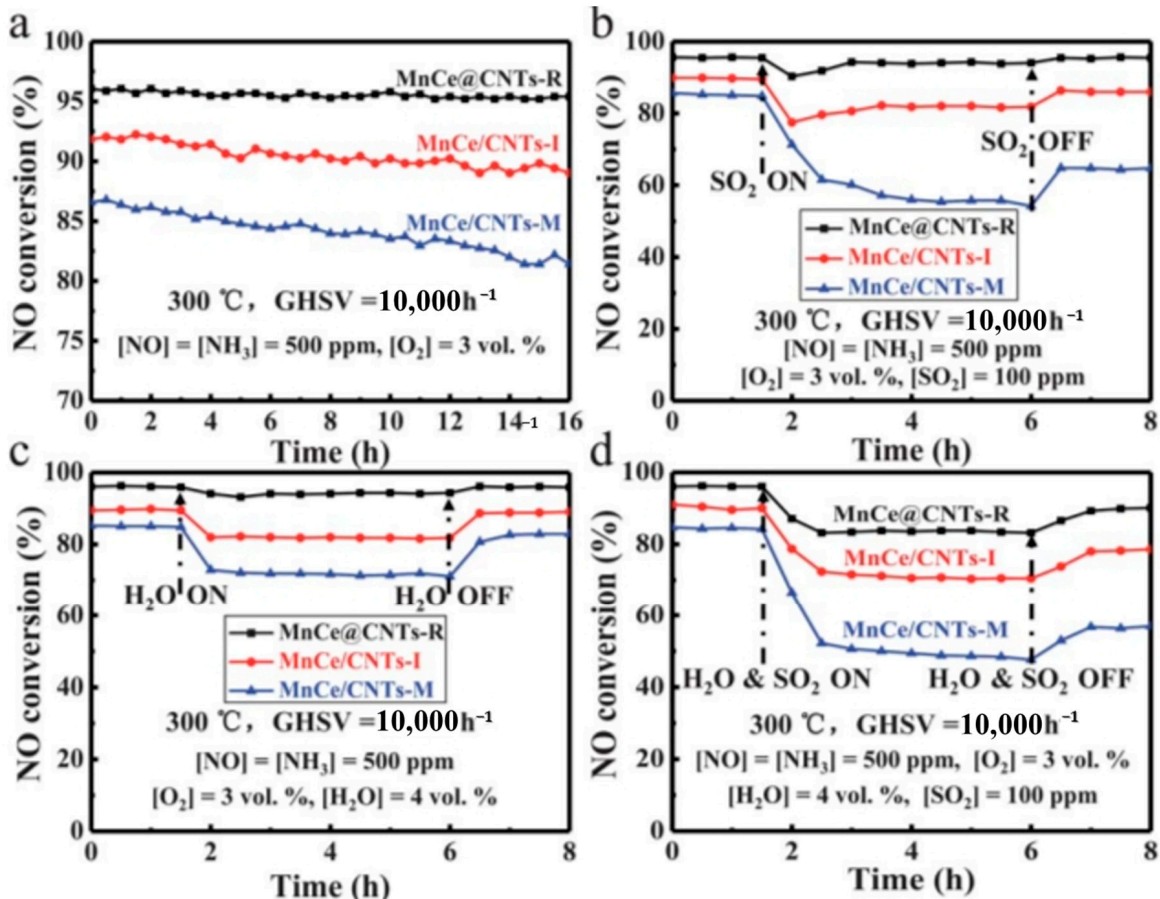

**Figure 15.** NH$_3$-SCR performance of Mn–Ce@CNTs–R catalysts. (**a**) Reaction conditions: [NO] = [NH$_3$] = 500 ppm, [O$_2$] = 3 vol%, N$_2$ balance, and GHSV = 10,000 h$^{-1}$. (**b**) Reaction conditions: [NO] = [NH$_3$] = 500 ppm, [O$_2$] = 3 vol%, N$_2$ balance, and GHSV = 10,000 h$^{-1}$, [SO$_2$] = 500 ppm. (**c**) Reaction conditions: [NO] = [NH$_3$] = 500 ppm, [O$_2$] = 3 vol%, N$_2$ balance, and GHSV = 10,000 h$^{-1}$, [H$_2$O] = 4 vol%. (**d**) Reaction conditions: [NO] = [NH$_3$] = 500 ppm, [O$_2$] = 3 vol%, N$_2$ balance, and GHSV = 10,000 h$^{-1}$, [H$_2$O] = 4 vol%, [SO$_2$] = 500 ppm. (Reproduced with permission from reference [90], Copyright 2018, Royal Society of Chemistry (London, UK)).

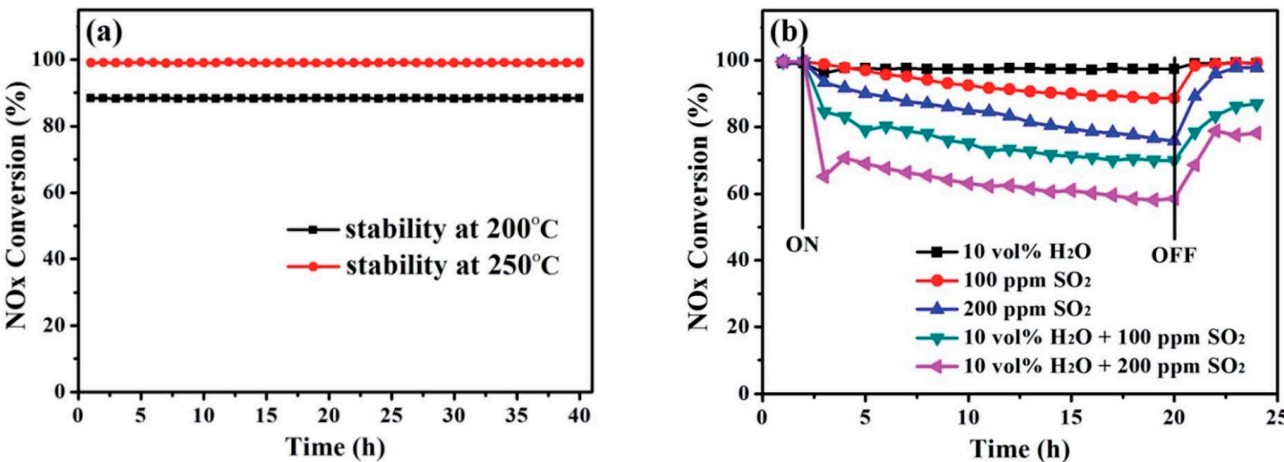

**Figure 16.** (**a**) The study of thermal stability and (**b**) H$_2$O resistance and SO$_2$ tolerance at 250 °C on CeMo(0.3) hollow microspheres. Reaction conditions: [NO] = [NH$_3$] = 500 ppm, [O$_2$] = 5%, balanced in N$_2$. (Reproduced with permission from reference [92], Copyright 2016, Royal Society of Chemistry (London, UK)).

### 3.2.3. Polymetallic-Modified Catalyst

Polymetallic-modified catalysts mainly introduce Fe–Mn–Ce catalysts, V–Mn–Ce catalysts, and other polymetallic-modified catalysts.

#### Fe–Mn–Ce Catalysts

Fe is often used as an additive to modify the catalyst. The effect of doping Fe on AC-supported Mn–Ce oxide catalysts are investigated for $NH_3$-SCR. When the content of Fe is 5%, the NO conversion of Mn–Ce–Fe/AC is 90% at 125 °C and 12,000 $h^{-1}$ space velocity [94]. As shown in Figure 17, the metal ions can enter the graphite crystal structure of AC and divide it into smaller graphene fragments. The doping of Fe can inhibit the decrease in surface area in the calcining process of the catalyst. In addition, the ratios of $Mn^{4+}/Mn^{n+}$ and $Ce^{3+}/Ce^{n+}$ and the amount of adsorbed oxygen and acid increase significantly after Fe doping. The main reason is that the Fe species expose the active sites of the acid or influence the chemical state of Mn/Ce species. The performance and $SO_2$ resistance of Mn–Fe–Ce/ACN catalysts is better than that of Mn/ACN catalysts. As shown in Figure 18, the surface acidity, reducibility, and surface chemisorbed oxygen are improved due to the addition of $FeO_x$ and $CeO_2$, which promote the $NH_3$-SCR performance [95]. In addition, the stronger surface acidity inhibits the adsorption of $SO_2$ and the consumption of $SO_2$ to adsorbed $NH_3$. Moreover, a small amount of $SO_2$ adsorbed on the catalyst surface reacts preferentially with $CeO_2$ to protect the main active components $MnO_x$ and $FeO_x$ from sulfation.

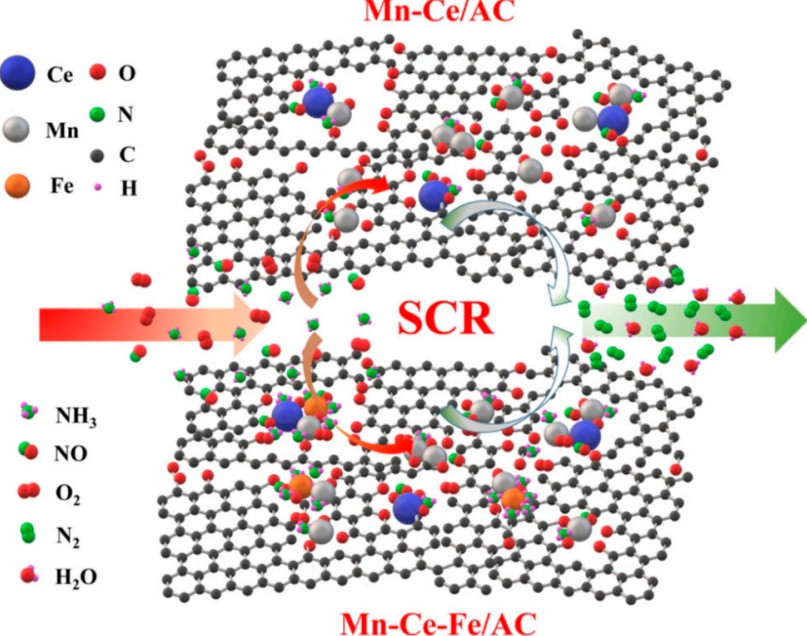

**Figure 17.** Reaction diagram of Mn–Ce/AC and Mn–Ce–Fe/AC catalysts for NO removing. (Reproduced with permission from reference [94], Copyright 2020, Elsevier (Amsterdam, The Netherlands)).

#### V–Mn–Ce Catalyst

For Mn–Ce/AC catalysts, doping $V_2O_5$ can obviously increase the NO conversion and improve $SO_2$ resistance. Whether it is V–Mn–Ce/AC or Mn–Ce/AC catalysts, their NO conversions are close to 98% at 200 °C. After adding $SO_2$, the NO conversion decreases slightly to about 90% for V–Mn–Ce/AC catalysts, and there is little change after $SO_2$ removal. For Mn–Ce/AC catalysts, the NO conversion drops significantly to around 68%, and it hardly restores after $SO_2$ removal. In the coexistence of $SO_2$ and $H_2O$, the deactivation of V–Mn–Ce/AC is more pronounced and the inhibition is irreversible, compared to the presence of $SO_2$ alone (Figure 19) [54]. Doping $V_2O_5$ enriches the chemically adsorbed oxygen and

enhances the surface acidity, which accelerates the reaction of SCR. To some extent, the $V_2O_5$ clusters prevent $SO_2$ forming and sulfating the dispersed Mn–Ce solid solution.

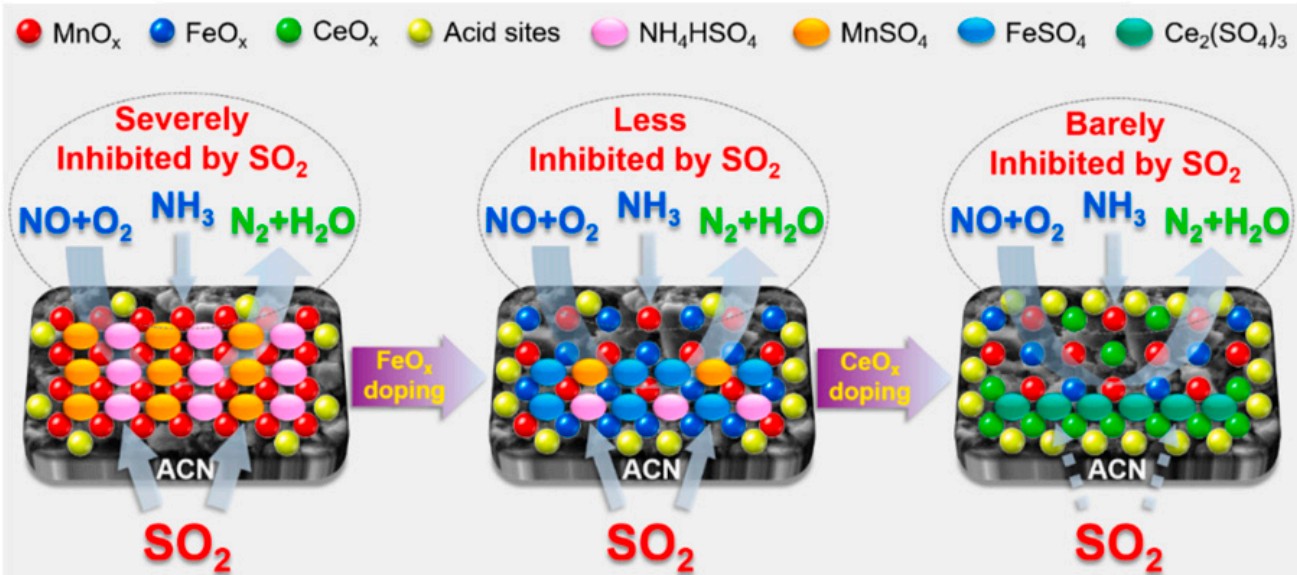

**Figure 18.** Mechanism of better $SO_2$ tolerance by doping $FeO_x$ and $CeO_x$ to Mn/CAN. (Reproduced with permission from reference [95], Copyright 2021, Elsevier (Amsterdam, The Netherlands)).

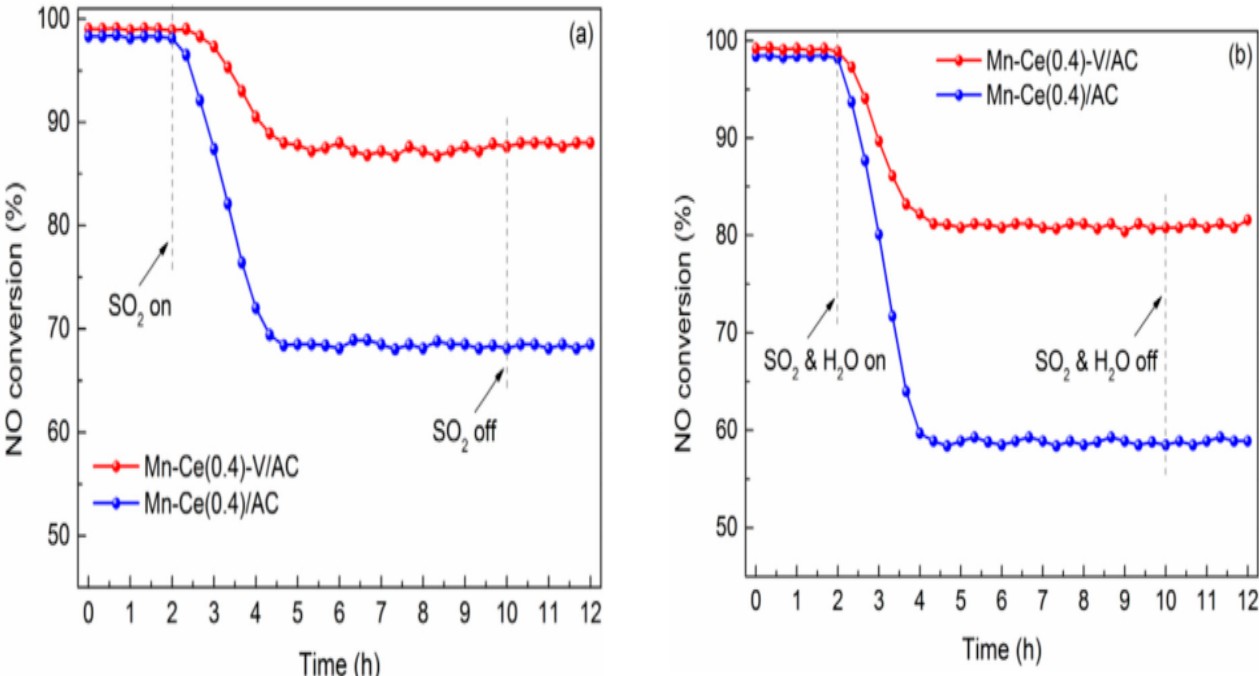

**Figure 19.** NO conversion over (**a**) Mn–Ce/AC and V–Mn–Ce/AC in the presence of $SO_2$ and (**b**) combined $SO_2/H_2O$ at 200 °C. (Reproduced with permission from reference [54], Copyright 2019, Elsevier (Amsterdam, The Netherlands)).

Other Polymetallic-Modified Catalysts

Novel core-shell structure $SiO_2@FeO_x$–$CeO_x$/CNTs catalysts have good $SO_2$ resistance and high stability. The introduction of 500 ppm $SO_2$ shows little influence on the catalytic activity, and the NO removal rate is about 90% [96]. Fe–$V_2O_5$/$TiO_2$-CNT catalysts also show excellent SCR performance. The NO conversion of the $Fe_3$-V/Ti–C catalyst decreases from 96% to 91% after the addition of $SO_2$ [97]. When $SO_2$ is stopped, the activity recovered to 94% (Figure 20). In the coexistence of oxygen, water vapor, and sulfur dioxide, the NO conversion of the Cu–Ce–Fe–Co/AC catalyst remains at 100% for 240 min [98].

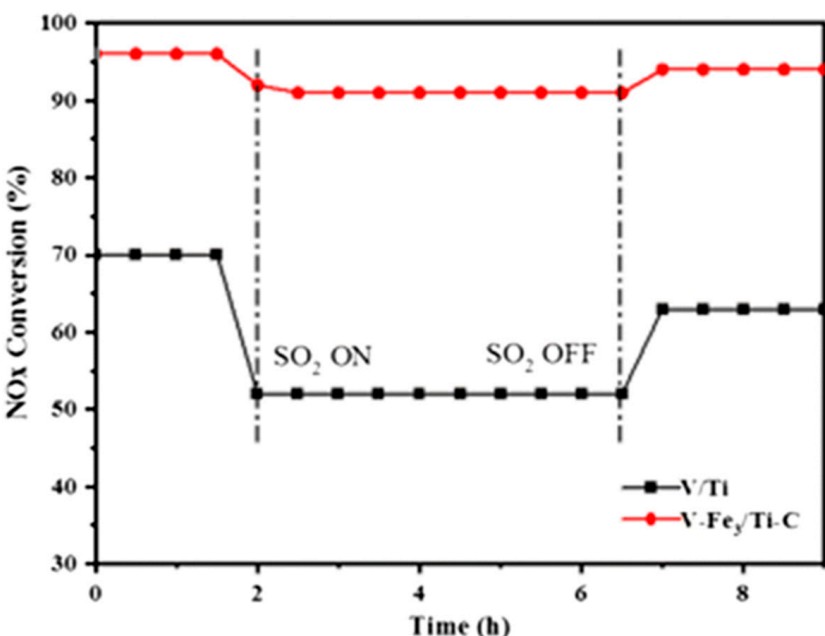

**Figure 20.** Catalytic reduction of $NO_x$ over V/Ti and V–$Fe_3$/Ti–C catalysts. (Reproduced with permission from reference [97], Copyright 2020, Springer Netherlands (Berlin, Germany)).

### 3.3. Application of Theoretical Calculation

As one of the most commonly used methods in computational chemistry, density functional theory (DFT) takes electron density instead of wave function as the research object. For DFT calculations [99–102], it can establish that quantity a SCR catalytic reaction is a fast process which cannot be understood by experimental characterization. The structure–activity relationships are revealed by simulating the adsorption process of active molecules and toxic molecules on the catalyst surface. The adsorption behavior of NO, $NH_3$, and $O_2$ on the $Fe_xO_y$/AC surface is investigated by DFT. On vacancies of the AC surface, $Fe_xO_y$ clusters are stably adsorbed with accompanying charge redistribution. These are nitroso, nitro, and nitrite when NO is adsorbed on the surface of $Fe_xO_y$/AC (Figure 21) [103]. Through DFT analysis and calculation, the adsorption process of reaction gases on the Mn–MOF-74 metal-organic framework catalyst system can be obtained. It was found that the molecules had competitive adsorption at the Mn metal sites. Compared with $H_2O$, $SO_2$ can displace $NH_3$ more easily, which explains the poisoning difference between $H_2O$ and $SO_2$ [104]. In addition, DFT calculations have shown that the SCR activity of the $Fe_xO_y$/AC catalyst is 28% higher than that of the corresponding oxide catalyst. Carbon deposition increases both the amount of intermediate/strong acid centers and the reducibility of catalytic centers.

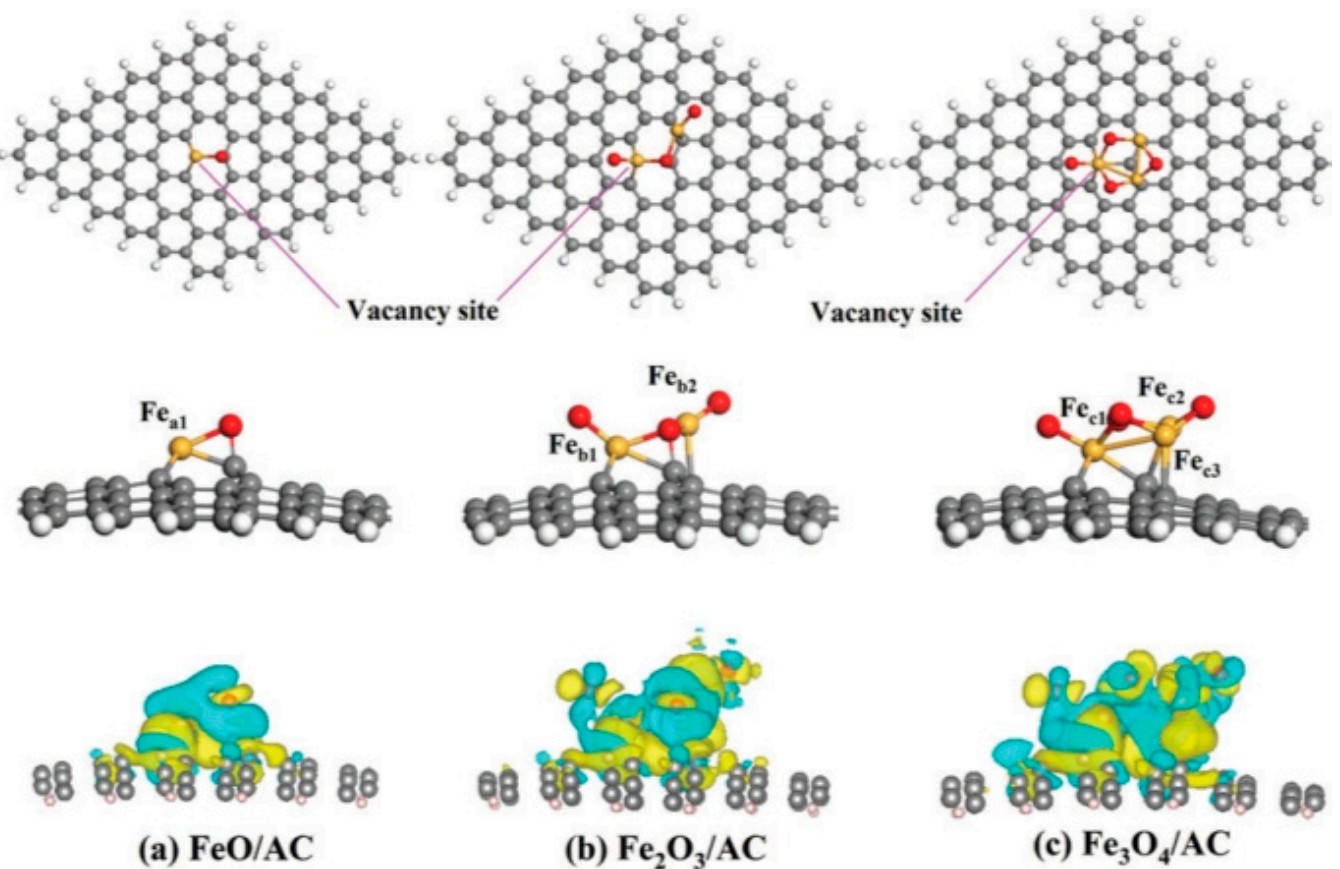

**Figure 21.** Configurations of Fe$_x$O$_y$/AC surface (Fe: golden, O: red, C: gray, H: white) and the electron density difference plots of Fe$_x$O$_y$ supported on the AC surface (cyan: lose electron, yellow: obtained electron). (Reproduced with permission from reference [103], Copyright 2021, Royal Society of Chemistry (London, UK)).

## 4. Other Strategies for Improving Sulfur and Water Resistance

In addition to selecting appropriate supports and active components for modification, it can also be improved by controlling the preparation method and reaction conditions of the catalyst for the SO$_2$ and H$_2$O resistance.

### 4.1. Preparation Methods

The successful support of metal oxide catalysts on adsorbents can be achieved by a variety of synthetic methods. So far, the preparation methods of denitration catalysts include the sol–gel method, citric acid method, precipitation method, ion exchange method, and impregnation method. The preparation method mainly affects the physical properties of the catalyst, including surface oxygen vacancy, pore volume, and specific surface area [105–107]. At present, the most common synthetic methods are impregnation and deposition precipitation. The enrichment of carbon support by completely immersing and mixing it with a metal precursor (i.e., copper nitrate solution) is called wet pore volume impregnation [108]. The MnO$_x$–CeO$_2$/P-CA catalysts were prepared via incipient wetness impregnation using Mn(NO$_3$)$_2$·4H$_2$O and Ce(NO$_3$)$_3$·6H$_2$O as precursors on phosphorus-doped carbon aerogels (P-CA) for NH$_3$-SCR of NO. The hydrophilicity of the carbon carrier for the MnO$_x$–CeO$_2$/P-CA catalyst was improved, which was conducive to the dispersity of active components and enhanced the electronic interaction between MnO$_x$ and CeO$_2$. Even with the presence of SO$_2$, the catalyst can still adsorb and oxidize more NO, forming more NO complexes. And the suppression of SO$_2$ to the SCR reaction is reduced through the L-H mechanism, which enhances the anti-SO$_2$ ability (Figure 22).

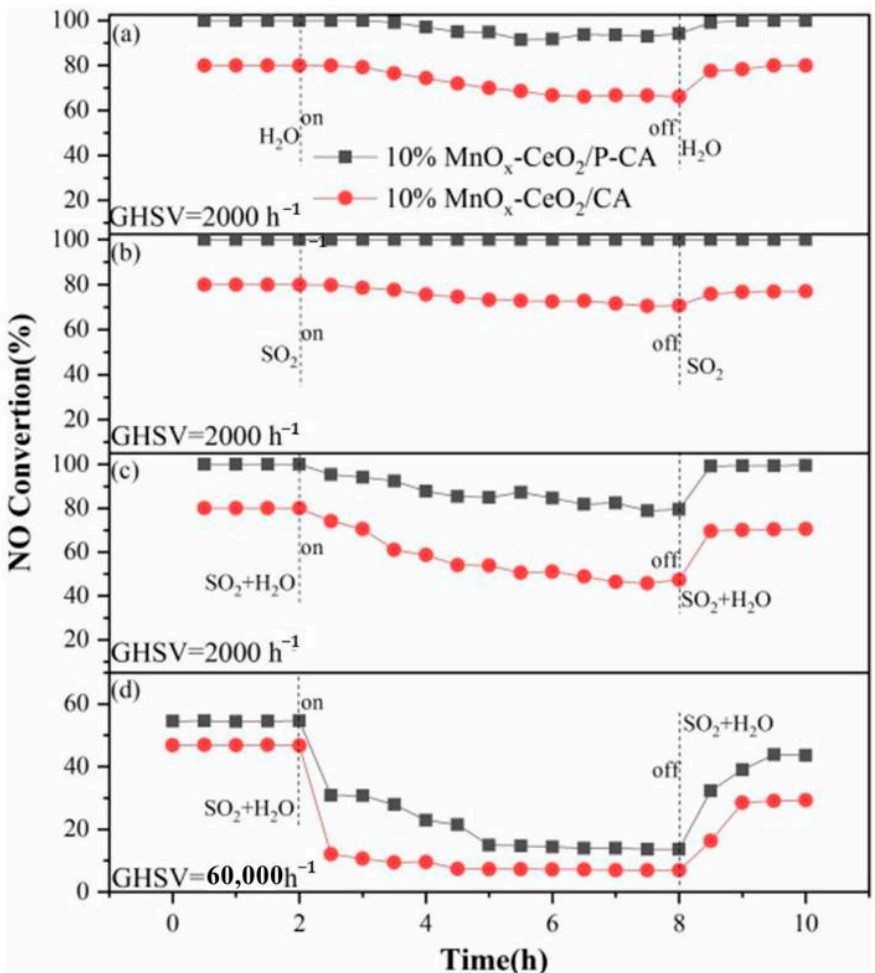

**Figure 22.** The influence of (**a**) $H_2O$, (**b**) $SO_2$, and (**c**) $SO_2$ + $H_2O$ on the NO conversion of 10%MnO$_x$–CeO$_2$/CA and 10%MnO$_x$–CeO$_2$/P–CA catalysts under GHSV of 2000 h$^{-1}$ at 160 °C, and (**d**) $SO_2$ + $H_2O$ resistance of 10%MnO$_x$–CeO$_2$/CA and 10%MnO$_x$ CeO$_2$/P-CA catalysts under GHSV of 60,000 h$^{-1}$ at 200 °C. (Reproduced with permission from reference [108], Copyright 2021, Wiley-Blackwell Publishing Ltd. (Weinheim, Germany)).

Deposition precipitation means that the metal precursor (i.e., nitric acid) is dissolved with the precipitant, and the carbon support (bulk) is completely mixed and heated. The precipitation method could make the distribution of active components more uniform so they have better adsorption and oxidation properties for NO. However, the precipitation method generates more impurities, which affects the performance of the catalyst. The operation of the impregnation method is simple, while the dispersity of the obtained catalyst's metal active particles is low.

### 4.2. Preparation and Reaction Conditions

The catalytic performance of catalysts is affected by reaction temperature, space velocity, initial NO concentration, $O_2$ concentration, $NH_3$ concentration, and $SO_2$ concentration [98]. The NO$_x$ conversion of Mn–Ce/AC catalysts calcined in a $N_2$, $O_2$, and air atmosphere is 94%, 75.6%, and 85.6%, respectively [109]. The catalyst calcined under the $N_2$ atmosphere could increase the dispersity of metallic oxides and enhance the surface acidity, reducing the oxidation and crystal formation of $MnO_2$ which positively impacts the catalytic oxidation performance and sulfur tolerance on catalysts. In addition, the impacts of loadings and precursors on the activity of MnO$_x$/ACs catalysts have been studied [110]. $Mn_3O_4$ using manganese acetate tetrahydrate as a precursor with a loading of 8% shows the best removal effect on NO, and the removal rate is 97% at 180 °C (Figure 23).

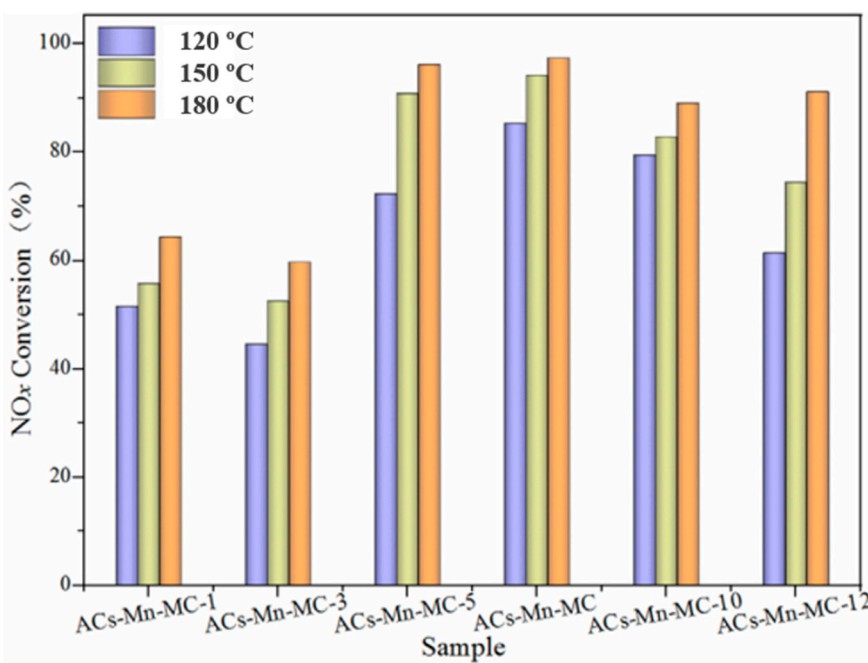

**Figure 23.** NO$_x$ conversion of samples with different loading amounts of Mn in the testing conditions: T = 120–180 °C, v(O$_2$) = 3.0%, c(NO) = c(NH$_3$) = 800 ppm, and GHSV = 10,000 h$^{-1}$ [110]. (Reproduced with permission from reference [110], Copyright 2020, Elsevier (Amsterdam, The Netherlands)).

## 5. Conclusions and Perspectives

Low-temperature SCR technology is one of the most effective technologies for removing NO$_x$ from flue gas. Carbon-based catalysts show excellent low-temperature activity and are a promising SCR catalyst. However, the presence of SO$_2$ and H$_2$O in the flue gas can easily lead to catalyst poisoning and deactivation. This paper reviews the research progress of low-temperature SCR for NO$_x$ removal against SO$_2$ and H$_2$O of carbon-based catalysts. The low-temperature NH$_3$-SCR reaction mechanism and SO$_2$ or/and H$_2$O poisoning mechanism are discussed, and the main strategies to enhance the SO$_2$ and H$_2$O resistance are summarized. In addition, the characteristics of common carbon-based materials are introduced. Moreover, the advantages and disadvantages of main low-temperature carbon-based SCR catalysts are evaluated comprehensively, and the application of DFT theoretical calculation in catalyst design and improvement of sulfur and water resistance is analyzed. Although great progress has been made in recent years, many catalysts can only be used under laboratory conditions and do not meet practical industrial application needs, so more in-depth research is needed.

(1) At present, the mixing or doping of metal oxides to carbon-based catalysts has been extensively studied to enhance H$_2$O and SO$_2$ resistance. However, the mechanism of anti-sulfur and anti-water reactions of carbon-based catalysts has not been thoroughly explored.

(2) When the carbon-based material is used as the carrier of the catalyst reaction, in addition to providing a larger specific surface area, the active site is increased and the activity is improved. In this process, whether the carbon-based materials participate in the reaction process or have a certain influence on the active components needs further study.

(3) Many studies have focused on the effect of supports on the activity of carbon-based catalysts and their resistance to sulfur and water. However, the influences of different preparation processes on the catalytic performance and sulfur resistance of carbon-based material have not been fully considered and studied. The influence of various factors on the anti-sulfur and water-resistance of carbon-based catalysts needs to be further studied.

(4) Mn–Ce carbon-based catalysts have excellent low-temperature $NH_3$-SCR activity and certain resistance to sulfur and wate, and are considered as the most promising denitrification catalysts. However, the research of Mn–Ce carbon-based catalysts is mainly in the laboratory stage, and its original poisoning mechanism needs to be further explored.

(5) In the actual industrial flue gas, in addition to $SO_2$ and $H_2O$, it also contains heavy metals, alkali metals, and dust. Therefore, the influence mechanisms of $SO_2/H_2O$ and other harmful substances on carbon-based catalysts need to be studied. Moreover, corresponding anti-poisoning strategies should also be proposed.

(6) Noble metal catalysts have excellent catalytic performance, but they cannot be used in large scale due to the high cost. The use of non-noble metal catalysts instead of noble metal has become the focus and general trend of research. However, most of the non-noble metal carbon-based catalysts possess poor stability and general sulfur and water resistance and are only used in the laboratory. It is necessary to develop non-precious metal carbon-based catalysts with excellent sulfur and water resistance under industrial conditions.

(7) Each industrial flue gas containing $NO_x$ has its own characteristics, and a single catalyst is difficult to meet the needs of $NO_x$ removal in all flue gases. Therefore, high-performance $NH_3$-SCR catalysts should be designed and developed according to the characteristics of the actual flue gas.

(8) At present, carbon-based SCR catalysts have achieved corresponding results in improving sulfur and water resistance, but most catalysts are still inactivated after use for a period of time. In my opinion, the rapid and low-cost regeneration methods of deactivated catalysts and corresponding mechanisms are another topic for future research.

**Author Contributions:** Z.S.: Conceptualization, writing the original draft, and formal analysis; S.R.: writing, review, and editing, funding acquisition, and supervision; B.Z.: conceptualization; W.B.: formal analysis and investigation; X.X.: writing, review, and editing, funding acquisition, and supervision; Z.Z.: review and editing. All authors have read and agreed to the published version of the manuscript.

**Funding:** This research was funded by the Natural Science Basic foundation of China (No. 52174325), the Shaanxi Province key research and development plan project (2019TSLGY05-05), the Shaanxi Province innovation ability support plan (2023-CX-TD-53), and the Shaanxi Provincial Department of Education Key Laboratory Scientific Research Project (20JS072). And The APC was funded by 20JS072. The authors gratefully acknowledge their support.

**Data Availability Statement:** Not applicable.

**Conflicts of Interest:** Authors Baoting Zhang and Weixin Bian were employed by the company Hanzhong Iron and Steel Co., Ltd. The remaining authors declare that the research was conducted in the absence of any commercial or financial relationships that could be construed as a potential conflict of interest. There are no conflict of interest.

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
