# Peer review of "Sulfur and Water Resistance of Carbon-Based Catalysts for Low-Temperature Selective Catalytic Reduction of NOx: A Review"

_catalysts, doi:10.3390/catal13111434_

Round 1

Reviewer 1 Report

Comments and Suggestions for Authors

Catalysts-2654400

Comments

The presented review article reports the  recent advances in the reaction mechanism of carbon-based catalysts (i.e., Activated carbon/coke, Activated carbon fiber, Graphene, carbon nanotube) for selective catalytic reduction of NOx with NH3 along with the poisoning mechanism of SO2 and H2O. . The review article could be accepted after addressing the following comments:

  1. The authors should emphasize the novelty of this review in the abstract and introduction. Also there are many recent reviews related to this area, so the authors could add a  focus table to show the novelty of this review compared with others. 
  2. The similarity rate is relatively high 38% according to iThenticate excluding references,, so it should be decreased to lower than 25 %
  3. The statement related to copyright permission should be added to the figures caption for example ‘’ Reproduced with permission from Ref. [x] Copyright year, Publisher  name’’
  4. The authors should provide qualitative and quantitative analysis on the discussed examples and also they should give their own opinion
  5. In the abstract  ‘’Due to the advantages of good adsorption performance, well-developed porous structure and large specific surface area, carbon-based materials have broad application prospects in support of catalysts for selective catalytic reduction of NOx with NH3 (NH3-SCR).’’ Rewrite. Also not all carbon materials have such merits, only porous carbon materials with great surface area and interconnected pores can enhance the adsorption of reactants, while integration of metal-based active sites can enhance the catalytic performance for reduction of NOx  
  6. In page 13 ‘’Fe is mainly dispersed in the form of γ-Fe2O3. Adding an appropriate amount of Mn increases the surface adsorption of oxygen Oβ and Fe3+, providing more active sites (Fig. 16).’’ XPS should be discussed in terms of metal phases Mo/M+ and their d-band level which are the main reason for promoting catalytic activity
  7. In NOx reduction, some important factors such as activation energy,  TOF/TON, and stability should be
  8. In ‘’Conclusion and perspectives’’, the authors should summarize the main findings in this review and also highlight the most important catalyst, most promising method, modifications, etc. Also, they should add their own suggestions supported by some citations
  9. The literature review is not enough, and more recent reviews related to carbon-based catalysts should be cited like 

https://doi.org/10.1039/D2GC02748H & https://doi.org/10.1016/j.envres.2022.112685 and others

Comments on the Quality of English Language

Polishing is required

Author Response

  1. The authors should emphasize the novelty of this review in the abstract and introduction. Also there are many recent reviews related to this area, so the authors could add a focus table to show the novelty of this review compared with others.

Response: Thanks to the reviewer for your advice. We have modified it in the revised manuscript.

‘…Notably, reviews on the sulfur and water resistance of carbon-based low-temperature NH3-SCR catalysts have not been previously reported, to the best of our knowledge.’…

Over time, the related scholars have conducted a lot of work on sulfur and water resistance of low-temperature NH3-SCR catalysts and achieved remarkable results. In addition, the reviews in this area have been widely reported. A review by Zhang et al. comprehensive overviewed the progress of Mn-based catalysts in sulfur and water resistance, and focused on analyzing the challenges and opportunities will facing in the development of Mn-based catalysts [37]. A review by Xu et al. provided the reaction mechanism of Ce-based catalysts for low-temperature NH3-SCR, and the technical to improve the resistance to sulfur and water were emphasized [38]. The recent review by Tang et al. also summarized the research progress of Mn-based catalysts in improving the denitrification activity of NH3-SCR at low temperature and the resistance to sulfur and water, and the challenges and possible solutions for designing catalyst systems with high sulfur and water resistance were discussed in detail [39]. However, the review of sulfur and water resistance of carbon-based catalyst for low-temperature NH3-SCR is rarely reported. This paper has reviewed the research findings on carbon-based catalysts in sulfur and water resistance in recent years.’…

  1. The similarity rate is relatively high 38% according to iThenticate excluding references, so it should be decreased to lower than 25%.

Response: Thank you very much for the suggestion. We have reduced the similarity rate in the revised manuscript.

  1. The statement related to copyright permission should be added to the figures caption for example ‘’Reproduced with permission from Ref.[x] Copyright year, Publisher name’’.

Response: Thank you very much for the suggestion. We have supplemented the manuscript with information about the references to the pictures.

  1. The authors should provide qualitative and quantitative analysis on the discussed examples and also they should give their own opinion.

Response: Thanks to the reviewer for your advice. We have made a qualitative and quantitative analysis of the examples given in the manuscript and presented our own views.

  1. In the abstract ‘Due to the advantages of good adsorption performance, well-developed porous structure and large specific surface area, carbon-based materials have broad application prospects in support of catalysts for selective catalytic reduction of NOx with NH3 (NH3-SCR).’ Rewrite. Also not all carbon materials have such merits, only porous carbon materials with great surface area and interconnected pores can enhance the adsorption of reactants, while integration of metal-based active sites can enhance the catalytic performance for reduction of NOx.

Response: Thanks for this comment. We have modified it in the revised manuscript.

‘…Low-temperature NH3-SCR is an efficient technology for NOx removal from flue gas. The carbon-based catalyst designed by using porous carbon material with great specific surface area and interconnected pores as the support to load the active components showed excellent NH3-SCR performance and has a broad application prospect.’

  1. In page 13 ‘’Fe is mainly dispersed in the form of γ-Fe2O3. Adding an appropriate amount of Mn increases the surface adsorption of oxygen Oβ and Fe3+, providing more active sites (Fig. 16).’’ XPS should be discussed in terms of metal phases Mo/M+ and their d-band level which are the main reason for promoting catalytic activity.

Response: Thank you very much for the suggestion. The authors have rediscussed this part in the manuscript based on metal phases Mo/M+ and their d-band level.

‘…The influences of SO2, H2O, or SO2+H2O on the NOx conversions of 8Mn6Fe/AC catalyst is investigated [85]. The NOx conversion decreases by 12% when H2O is introduced into the reactor due to a competitive adsorption between NH3 and H2O. And then the activity remains stable. Fe is mainly dispersed in the form of γ-Fe2O3. After the introduction of Mn, the ratio of Fe3+/Fe2++Fe3+ changed little, and the value of Oβ/(Oβ+Oα) increased significantly. The doping of Mn increases the amount of chemisorbed oxygen with higher migration ability, which could oxidize NO to NO2 to form “fast SCR” reaction, thus promoting the denitration performance and SO2 resistance of catalysts (Figure 14).’…

  1. In NOx reduction, some important factors such as activation energy, TOF/TON, and stability should be.

Response: Thanks for this comment. We have reconsidered these important factors and discussed them in detail in the revised manuscript.

  1. In ‘’Conclusion and perspectives’’, the authors should summarize the main findings in this review and also highlight the most important catalyst, most promising method, modifications, etc. Also, they should add their own suggestions supported by some citations.

Response: Thanks to the reviewer for your advice. We have summarized the main findings in this review and also highlighted the most important catalyst, the most promising method, and modifications, etc. And the suggestions have been added to the ‘Conclusion and perspectives’.

Low temperature SCR technology is one of the most effective technologies to remove NOx from flue gas. Carbon-based catalysts show excellent low temperature activity and are a promising SCR catalyst. However, the presence of SO2 and H2O in the flue gas can easily lead to catalyst poisoning and deactivation. This paper reviews the research progress of low-temperature SCR for NOx removal against SO2 and H2O of carbon-based catalysts. The low-temperature NH3-SCR reaction mechanism and SO2 or/and H2O poisoning mechanism are discussed, the main strategies to enhance the SO2 and H2O resistance are summarized. In addition, the characteristics of common carbon-based materials are introduced. Moreover, the advantages and disadvantages of main low temperature carbon-based SCR catalysts are evaluated comprehensively, and the application of DFT theoretical calculation in catalyst design and improvement of sulfur and water resistance is analyzed. Although great progress has been made in recent years, many catalysts can only be used under laboratory conditions and do not meet practical industrial application, so more in-depth research is needed.…’

(4) The carbon-based materials supported by Mn-Ce catalysts have high low temperature NH3-SCR activity and certain resistance to sulfur and water, and are considered as the most promising denitrification catalysts. However, the research of Mn-Ce carbon-based catalyst is mainly in the laboratory stage, and its original poisoning mechanism needs to be further explored.

(5) In the actual industrial flue gas, in addition to SO2 and H2O, it also contains heavy metals, alkali metals and dust. Therefore, the influence mechanism of SO2/H2O and other harmful substances on carbon-based catalysts needs to be studied. Moreover, corresponding anti-poisoning strategies should also be proposed.

(7) Each industrial flue gas containing NOx has its own characteristics, and a single catalyst is difficult to meet the needs of NOx removal in all flue gases. Therefore, high performance NH3-SCR catalysts should be designed and developed according to the characteristics of actual flue gas.

(8) At present, carbon-based SCR catalysts have achieved corresponding results in improving sulfur and water resistance, but most catalysts still inactivation after use for a period of time. In my opinion, the rapid and low-cost regeneration methods of deactivated catalysts and corresponding mechanisms are another topic for future research.

  1. The literature review is not enough, and more recent reviews related to carbon-based catalysts should be cited like. https://doi.org/10.1039/D2GC02748H&https://doi.org/10.1016/j.envres.2022.112685 and others.

Response: Thanks for this comment. We have read the relevant literatures in detail and quoted them in the revised manuscript.

‘67. Eid, K.; Gamal, A.; Abdullah, A.M. Graphitic carbon nitride-based nanostructures as emergent catalysts for carbon monoxide (CO) oxidation. Green Chem. 2023, 25, 1276-1310.

‘68. Nemati, F.; Rezaie, M.; Tabesh, H.; Eid, K.; Xu, G.B.; Ganjali, M.R.; Hosseini, M.; Karaman, C.; Erk, N.; Show, P.L. Cerium functionalized graphene nano-structures and their applications; A review. Environ. Res. 2022, 208, 112685.

Reviewer 2 Report

Comments and Suggestions for Authors

The topic covered in the paper, the sulfur and water resistance of Carbon-based catalysts for Low-T SCR of NOx, is certainly of interest. Unfortunately, this paper needs to be extensively improved.

In particular chapter 2 (2. Reaction mechanisms of carbon-based catalysts for SCR denitration) and all its subparagraphs, and some parts of chapter 3 ( i.e. 33.1.1 and 3.2.3). In these paragraphs, a large number of images derived from the literature relating to particular systems are reported, without a sufficient and clear explanation in the text of what they represent, and without a related discussion to make them representative of universal or particular concepts in the context of the research field of the review (which should be the aim of a review). For each figure, we find only 2-3 lines of incomplete and unclear description or discussion in the text. Furthermore, the comments in the text relating to Figure 3 and Fig 5 do not seem related to the figures.

Furthermore, in these chapters, the topics are often described insufficiently. They should be improved.

As an example of some major revisions to do, (but all the paragraphs have to be improved):

Fig 2 and Fig. 4  are very complex and the corresponding comments are non-sufficient. The discussion has to be improved.

Fig 3 The comment in the text does not correspond to the figure.

Fig 5 The comment in the text ( about H2O and NH3/NO) does not correlate with Figure 5 (related to H2O and SO2). Fig 5: the graphical quality is low. In particular section ( C ) is not readable.

Fig. 7: The meaning of the image is not commented on at all.

Fig 8: the comment is not sufficient.

Fig 9 and Fig 10: related discussion should be improved.

Fig 19 and Fig 20:  related discussion should be improved.

Author Response

Dear Reviews,

We are very grateful to you and reviewers of the paper for the critical reading of the manuscript and the recommendations for our further improvements. We have checked the manuscript and revised it according to the comments, and the major revised portions were highlighted. We also responded point by point to the reviewer’s comments as listed below.

Comments 2

  1. In particular chapter 2 (2. Reaction mechanisms of carbon-based catalysts for SCR denitration) and all its subparagraphs, and some parts of chapter 3 (i.e. 33.1.1 and 3.2.3). In these paragraphs, a large number of images derived from the literature relating to particular systems are reported, without a sufficient and clear explanation in the text of what they represent, and without a related discussion to make them representative of universal or particular concepts in the context of the research field of the review (which should be the aim of a review). For each figure, we find only 2-3 lines of incomplete and unclear description or discussion in the text. Furthermore, the comments in the text relating to Figure 3 and Fig 5 do not seem related to the figures. Furthermore, in these chapters, the topics are often described insufficiently. They should be improved.

Response: Thank you very much for the suggestion. We have given a full and clear explanation of the corresponding images in the revised manuscript and discussed the special and universal concepts they represent. In addition, the annotation of the graph has been modified accordingly.

  1. As an example of some major revisions to do, (but all the paragraphs have to be improved): Fig 2 and Fig. 4 are very complex and the corresponding comments are non-sufficient. The discussion has to be improved.

Response: Thanks to the reviewer for your advice. We have modified it in the revised manuscript.

‘…For CeO2-WO3/TiO2 Catalysts Figure 2, the introduction of SO2 prevents the generation of active intermediates, such as NH2NO. In addition, SO2 reacts with NH3 or CeO2 to formation (NH4)2SO4 and Ce2(SO4)3, respectively, which covers the active sites and inhibits the redox performance. This results in the irreversible inactivation. Besides, industrial flue gas usually contains other metals such as Cd, and SO2 binding with it also produces CdSO4 to cover the active sites, causing further deactivation of the catalyst. The production of sulfate on carbon-based catalysts surface covering the active sites and the vulcanization of the active metal are the two main forms of permanent deactivation. As shown in Figure 3, the existence of SO2 to MnFeOx catalyst results in the production of MnSO4, FeSO4 and (NH4)2SO4, which inhibits both L-H and E-R mechanism. Thus, the catalysts display poor SCR activity. When the Sm is dropped, SO2 is preferred combined with it. As a result, the active sites could be protected. The addition other metal that preferentially reacts with SO2 is the main method to enhance the SO2 resistance of carbon-based NH3-SCR catalysts.’

  1. Fig. 3 The comment in the text does not correspond to the figure.

Response: Thanks for this comment. After careful consideration and modification, we found that the content shown in Fig. 2 and Fig. 4 previously contained the information in Fig. 3, so we removed Fig. 3 from the manuscript.

  1. Fig. 5 The comment in the text (about H2O and NH3/NO) does not correlate with Figure 5 (related to H2O and SO2). Fig 5: the graphical quality is low. In particular section (C) is not readable.

Response: Thanks for this comment. After careful examination, Figure 5 is not appropriate here, and the information contained in Figure 6 fully supports the discussion in the text, we have removed it from the original manuscript.

  1. Fig. 7: The meaning of the image is not commented on at all.

Response: Thanks for this comment. The content contained in ‘Figure 7’ has been discussed at length in the revised manuscript.

‘…As shown in Fig. 5, when the SO2 is introduced, the SO2 is absorbed on MnOx and oxidized to SO3. When the sulphates accumulate to certain amount, the formation of SO42- polymer results in the reduction of the surface area and the inhibition of redox property. Washing with water can remove SO42- polymer and restore the catalytic activity of active components.’…

  1. Fig 8: the comment is not sufficient.

Response: Thanks for this comment. We have made sufficient comments on ‘Figure 8’ in the revised manuscript.

‘…T As shown in Figure. 6, the reaction of NO on MnOx-Cu/AC catalyst follows both L-H mechanism and E-R mechanism. For the E-R mechanism, NO in the gas phase reacts with NH3 activated by the acidic sites on the catalyst surface. For the L-H mechanism, NO and O2 in the gas phase interact and are oxidized to nitrates or nitrites intermediates by the catalyst, which then react with activated NH3. It is worth noting that regardless of the mechanism, the activation of NH3 at the acidic site is a key step for the reaction to proceed.’…

  1. Fig 9 and Fig 10: related discussion should be improved.

Response: Thank you very much for the suggestion. We have improved the related discussion of ‘Fig. 9’ and ‘Fig. 10’ in the revised manuscript.

‘…As shown in Fig. 7, after the introduction of SO2, the activity of V2O5/AC catalyst decreased rapidly, then had a weak rise, and finally decline slowly to remain a stable value[58]. However, when the SO2 poisoned V2O5/AC catalyst is characterized, it is observed that only a small part of the pores on surface are blocked by sulfate. The generation of VOSO4 by the reaction of V2O5 and SO2 will cause the NO conversion to drop sharply, which is the main reason for the deactivation of the catalyst. The sulfurization of active metals by SO2 was the main cause of irreversible inactivation of most carbon-based catalysts in sulfur-containing flue gas.’…

‘…For Mn-Ce/AC catalysts shown in Fig. 8, SO2 reacted with NH3 to inhibit the reaction of NO and NH3[55]. On the other hand, SO2 combined with manganese oxide and cerium oxide to form MnSO4 and (Ce)2(SO4)3, which caused permanent inactivation. The addition of V could improve the acidity of the catalyst surface and inhibit the combination of SO2 and NH3. In addition, the vulcanization of Mn-Ce solid solution is also prevented. Therefore, the sulfur resistance of the catalyst can be enhanced.’

  1. Fig 19 and Fig 20: related discussion should be improved.

Response: Thanks to the reviewer for your advice. We have improved the related discussion of ‘Fig. 19’ and ‘Fig. 20’ in the revised manuscript.

‘…The effect of doping Fe on AC supported Mn-Ce oxide catalysts are investigated for NH3-SCR. When the content of Fe is 5%, the NO conversion of Mn-Ce-Fe/AC is 90% at 125°C and 12000 h-1 space velocity [99]. As shown in Figure 17, the metal ions can enter the graphite crystal structure of AC and divide it into smaller graphene fragments. The doping of Fe can inhibit the decrease of surface area in the calcining process of catalyst. Besides, the ratios of Mn4+/Mnn+, Ce3+/Cen+, and the amount of adsorbed oxygen, acid increase significantly after Fe doping. The main reason is that the Fe species expose the active sites of the acid or influence the chemical state of Mn/Ce species.’…

‘…The performance and SO2 resistance of Mn-Fe-Ce/ACN catalyst is better than that of Mn/ACN. As shown in Figure 18, the surface acidity, reducibility and surface chemisorbed oxygen are improved due to the addition of FeOx and CeOx, which increase the NH3-SCR performance [100]. In addition, the stronger surface acidity inhibits the adsorption of SO2 and the consumption of SO2 to adsorbed NH3. Moreover, a small amount of SO2 adsorbed on the catalyst surface reacts preferentially with CeO2 to protect the main active components MnOx and FeOx from sulfation.’…

 Thank you again for reviewing the manuscript and made the valuable suggestions. The above was my modification to the paper. If you have any questions, please don't hesitate to contact us, and we will try our best to revise the manuscript.

With kind regards,

Shan Ren

Email: shan.ren@cqu.edu.cn

Round 2

Reviewer 1 Report

Comments and Suggestions for Authors

The authors replied to all comments, and the article could be accepted.

Reviewer 2 Report

Comments and Suggestions for Authors

The authors responded to all reviewers' comments by modifying the text according to the suggested indications. The paper is now improved, and acceptable for publication.